# Coevolving residues distant from the ligand binding site are involved in GAF domain function
Wesam S. Ahmed [1], Anupriya M. Geethakumari [1], Asfia Sultana[1], Anmol Tiwari[2], Tausif Altamash[1,4], Najla Arshad[3,5], Sandhya S. Visweswariah [2] & Kabir H. Biswas [1] ✉

Ligand binding to GAF domains regulates the activity of associated catalytic domains in various proteins, such as the cGMP-hydrolyzing catalytic domain of phosphodiesterase 5 (PDE5) activated by cGMP binding to GAFa domain. However, the specific residues involved and the mechanism of GAF domain function remain unclear. Here, we combine computational and experimental approaches to demonstrate that two highly coevolving residues, L267 and F295, distant from the ligand binding site, play a critical role in GAF domain allostery. Statistical Coupling Analysis (SCA) of GAF domain sequences identified these residues, and molecular dynamics (MD) simulations of both apo and holo forms of wild-type and mutant (L267A, F295A) PDE5 GAFa domains revealed significant changes in structural dynamics and cGMP interaction. Mutational incorporation into a Bioluminescence Resonance Energy Transfer (BRET)-based biosensors, which detects ligand-induced conformational changes, showed altered GAF domain conformation and increased $EC_{50}$ for cGMP-induced conformational changes. Similar effects were observed in full-length PDE5 and the GAF domain fluorescent protein, miRFP670nano3. Structural analysis of conformers observed in MD simulations suggested a mechanism by which these coevolving residues influence GAF domain allostery. Our findings provide insight into the role of distant residues in GAF domain function and may enhance understanding of allostery in proteins.

The activity of natural proteins, such as enzymes, is controlled by associated regulatory domains, which, in many cases, is achieved by binding a small molecule ligand and relaying associated structural changes to the catalytic domains. While many such regulatory domains are known, GAF domains (c**G**MP-specific PDEs, bacterial **A**denylyl cyclases, and bacterial **F**hLA transcriptional regulators) represent one of the largest families of regulatory domains[1], and are conserved in organisms ranging from archaea to chordates, including humans[2]. Indeed, GAF domains play a crucial role in regulating protein function that impacts a range of biological processes, including gene expression regulation in bacteria[3], light-detection and signaling in plant and cyanobacteria[4], ethylene detection and signaling in plants[5], nitrogen fixation in bacteria[6], cAMP binding feedback control in cyanobacterial adenylyl cyclase[7], and the two-component sensor histidine kinase in both bacteria and plants[8,9], as well as some of the mammalian cyclic

nucleotide phosphodiesterases (PDEs)[1,3,10]. While some GAF domains are found in tandem (e.g. the GAF domains of mammalian PDEs)[11], others are not (e.g. the GAF domain of cyanobacteriochrome photoreceptors and Anabaena Adenylyl cyclase)[7,12].

One of the well-studied GAF domains is the cGMP-binding GAFa domain in PDE5, which regulates cGMP levels in several human tissues, including vascular smooth muscle cells, lung, brain, kidney, cardiac myocytes, gastrointestinal tissue, platelets, and penile corpus cavernosum[11,13]. The N-terminally located GAFa domain in PDE5 binds cGMP with a relatively high affinity and specificity and is thought to induce a conformational change in the domain that is transduced to the C-terminally located, cGMP-hydrolyzing catalytic domain, thereby activating it and increasing cGMP hydrolysis[14,15]. Previous studies have shown that the apo GAF domain has high flexibility and that the

[1]College of Health & Life Sciences, Hamad Bin Khalifa University, Doha, Qatar. [2]Department of Developmental Biology and Genetics, Indian Institute of Science, Bengaluru, India. [3]Department of Immunobiology, Yale University School of Medicine, New Haven, CT, US. [4]Present address: Materials Science and Nano-Engineering (MSN) Department, Mohammed VI Polytechnic University (UM6P), Ben Guerir, Morocco. [5]Present address: Center for Cancer Cell Biology, Immunology, and Infection, Rosalind Franklin University of Medicine and Science, Chicago Medical School, North Chicago, IL, US.
✉e-mail: kbiswas@hbku.edu.qa

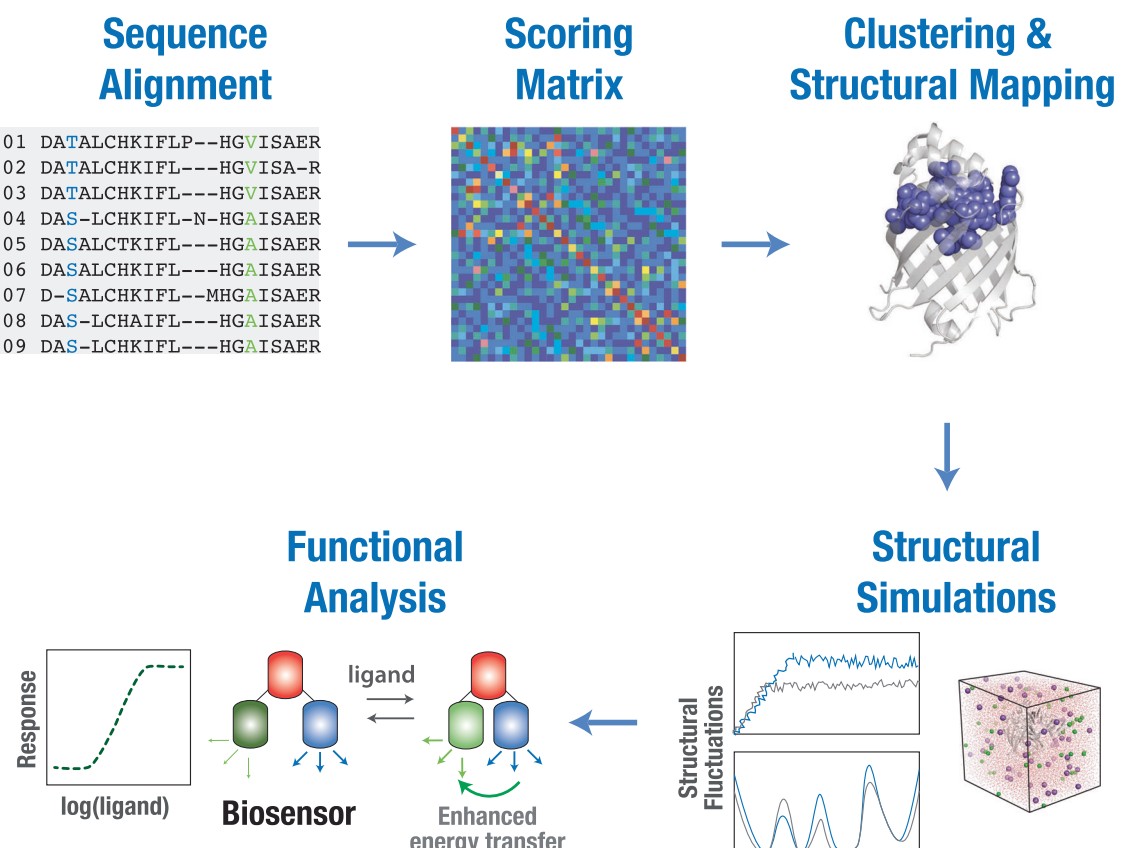

**Fig. 1 | Coevolving residue positions and their role in GAF domain allostery.**
Schematic showing an approach to determine coevolving residues and delineating their role in the allosteric regulation of GAF domains. GAF domain sequences were aligned, and statistical coupling analysis (SCA) was performed to detect coevolving residue positions. The scoring matrix obtained from SCA was then used for clustering to determine the cluster of coevolving residues with highest scores, which were then mapped on to a GAF domain structure. MD simulations followed by biophysical assays with biosensors were then used to determine the role of individual coevolving residue positions in the allosteric regulation of GAF domains.

conformational changes accompanying cNMP binding stabilize secondary structural elements in the domain[16,17]. This converts the GAF domain from an "open" to a "closed" conformation wherein the cNMP is deeply buried inside the ligand binding site. Overall, there is still a limited understanding of the complexity of this allosteric communication induced by cNMP binding to the GAF domain, including the identity of amino acid residues that are involved in the process.

In the current study, we combined protein sequence-based coevolutionary analysis, molecular dynamics (MD) simulations, and biophysical assays, to better understand ligand binding and allosteric regulation in GAF domains (Fig. 1). Specifically, we performed Statistical Coupling Analysis (SCA)[18,19] of GAF domain sequences to identify coevolving residue positions in the domain, that may play a role in ligand binding and regulatory function, and identified nine residue positions that showed high statistical coupling scores. Mapping these positions on the GAFa domain of PDE5 revealed two residues, namely L267 and F295, that showed high statistical coupling scores and were located distant from the cGMP binding site. MD simulation analysis of the wild-type (WT) and mutant (both apo and cGMP-bound, holo, forms) PDE5 GAFa domains revealed distinct changes in the structural dynamics of L267A and F295A mutants. Functional characterization using Bioluminescence Resonance Energy Transfer (BRET)-based biosensors revealed a structural change in the basal state and an increase in the $EC_{50}$ values of cGMP-induced conformational change in the mutant GAF domains (both isolated GAFa domain as well as in the full-length PDE5 protein). Moreover, genomic database and structure-based analysis predicted

that variations in these two positions in PDE5A gene were deleterious. Importantly, equivalent mutations in the fluorescent GAF domain, miRFP670nano3, resulted in a decrease in the fluorescence of the protein indicative of a conserved role of these two distant, coevolving residues in GAF domain function.

## Results

### Identification of coevolving residues in GAF domains using SCA
To determine coevolving residues and better understand the allosteric communication in the GAF domain, we used the statistical coupling analysis (SCA) coevolutionary analysis method followed by MD simulations and functional analysis to determine the role of the identified coevolving residues in GAF domain allostery (Fig. 1). SCA involves analysis of covariation between pairs of amino acid positions in multiple sequence alignments (MSA) to identify coevolving residues in a protein family[18,19]. It specifically measures the extent to which the distribution of amino acids at one position (x) is altered by changes in the amino acid distribution at another position (y). Towards this, GAF domain sequence alignment obtained from the Pfam database[20] was refined for phylogenetic relatedness and gaps, and SCA was performed using previously described algorithms[19,21] to generate a matrix of pairwise correlation scores for each residue in the GAF domain (Fig. 2A). Importantly, only ~1% of the residue pairs showed correlation scores >0.25, and hierarchical clustering analysis revealed a cluster of nine highly coevolving residue positions (Supplementary Fig. 1), suggestive of their role in ligand binding and allostery in GAF domains.

We mapped the cluster of coevolving residues on the cGMP-bound structure of the GAFa domain of PDE5 (amino acid residues 164–312;

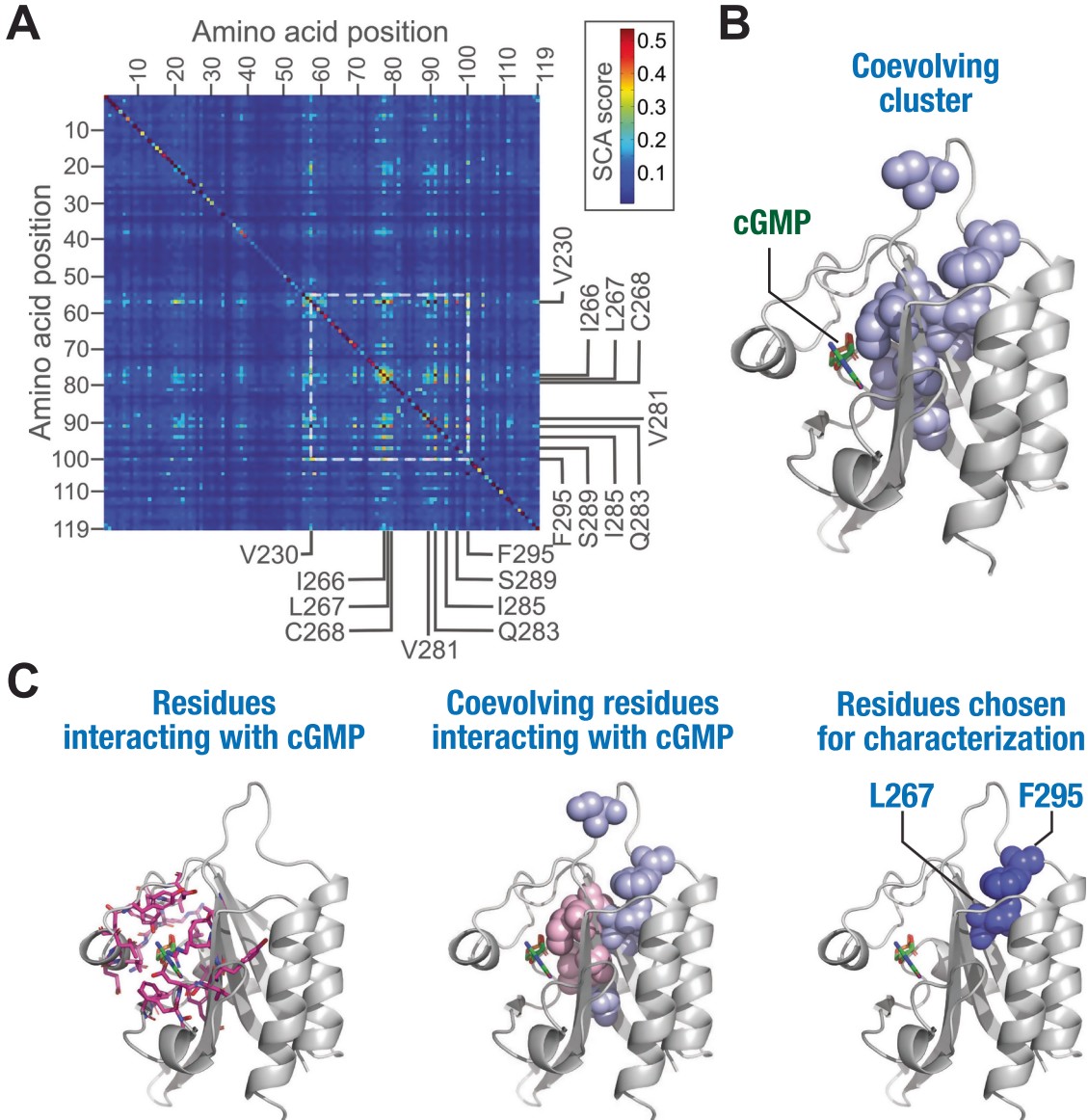

**Fig. 2 | SCA reveals a cluster of coevolving residues in GAF domains. A** Color-coded heatmap showing pairwise scores of individual residue positions in GAF domains obtained from SCA. The positions of the nine coevolving residues are indicated. The area in the heatmap where the coevolving residue positions intersect is highlighted by a white box. **B** Cartoon representation of the GAFa domain of PDE5 (PDB: 2K31[22]) showing the coevolving cluster of residues (spheres in light blue) in the GAF domains. **C** Cartoon representation of the GAFa domain of PDE5 showing residues (side chains as sticks in pink) within a 5 Å distance from cGMP (left panel), coevolving residues (spheres in light pink; rest of the coevolving residues in light blue) within 5 Å distance from cGMP (middle panel) and coevolving residues L267 and F295 selected for further functional characterization (dark blue spheres, right panel).

human PDE5A1 numbering; PDB: 2K31[22]) (Fig. 2B). We chose to use the GAFa domain of PDE5 for the following reasons: first, unlike other GAF domains that are either not known to bind small molecules or whose function is still unknown, the PDE5A GAFa domain has a well-known function, which is binding cGMP with high specificity and affinity, and allosterically activating the cGMP-hydrolyzing activity of PDE5 catalytic domain. The isolated PDE5 GAFa domain has been reported to show a thousand-fold higher affinity for cGMP compared to cAMP[22]. This selectivity is much higher compared to some other cGMP-binding GAF domains, such as the cGMP-binding GAFb domain of PDE2A, which shows only about 20-fold higher affinity for cGMP[23]. Second, the three-dimensional structures of the isolated PDE5 GAFa domain, both in the absence of cGMP (unliganded; apo)[16] and in the cGMP-bound (holo)[22] form, are available in the literature, which allows for robust structural comparison and subsequent computational investigations. Third, we have

previously engineered BRET-based conformational biosensors for both the isolated GAFa domain and the full-length PDE5. These biosensors faithfully report conformational changes in the proteins that occur as a result of mutation (basal BRET) or as a consequence of ligand-binding (cGMP-binding induced BRET), which has enabled the characterization of several mutations with regards to GAF domain allostery[24–26].

In the human PDE5, the GAFa domain is composed of six antiparallel β sheets (β1-β6) and five α helices (α1-α5) with an arrangement of the secondary structural elements of ααββββαβαββα[16]. Positional mapping of the nine coevolving residue positions (Fig. 2B, Supplementary Table 1) revealed that six residues (V230, I266, C268, V281, Q283, I285; all residue numbers indicated are as per human PDE5A1 sequence) are located in (V230, I266, Q283, I285) or near (C268, V281) the ligand binding site in the GAFa domain (binding site residues were identified using a cutoff of 5 Å[27] distance from cGMP in the holo GAFa domain[22]). On the other hand, the remaining

three residues (S289, L267, and F295) are located distant from the binding site with no direct interaction with the cGMP (Fig. 2C, Supplementary Movie 1). Interestingly, two of the distantly located residues (L267 and F295) had high coupling scores among the nine coevolving residues identified from SCA. A closer inspection of these two residues revealed that they form a hydrophobic interaction (Supplementary Movies 2, 3). Therefore, among the nine coevolving residues, we chose to further characterize the L267 and F295 residue positions individually in the PDE5 GAFa domain.

## MD simulations reveal distinct changes in the structural dynamics of the PDE5 GAFa domain upon mutation of coevolving residues distant to the cGMP binding site

To evaluate the structural and functional impact of the two highly coevolving positions (L267 and F295), we individually generated L267A and F295A mutations in the apo and holo structural models of the GAFa domain. We then performed three independent, all-atom, explicit solvent, 1000 ns-long, MD simulation runs of the WT and mutant GAFa domains in both the apo and holo forms. MD simulation has found several applications, including probing the effect of single-point mutations on the structure of proteins[28] and the determination of ligand binding affinity[27]. A detailed analysis of the MD simulation trajectories revealed distinct features in the structural dynamics of the apo and holo GAFa domains, as well as notable differences between the WT and the two mutant GAFa domains (Supplementary Movies 4 to 21). Comparison of the WT apo and holo forms of the GAFa domain confirmed previous findings that the apo form exhibits a greater flexibility to accommodate ligand binding and subsequent conformational changes (Fig. 3A). We observed the following trends in the root mean square deviation (RMSD) measurements. Specifically, analysis of the apo and holo WT GAFa domain trajectories revealed a reduced RMSD value in the cGMP-bound GAFa domain (4.5 ± 0.9 and 3.4 ± 0.7 Å for the apo and holo structures, respectively). Interestingly, while the apo-form of the L267A mutant showed a decrease (4.2 ± 0.7 Å) in RMSD, the F295A mutant apo-form showed an increase (4.5 ± 0.9) as compared to the WT GAFa domain (4.9 ± 0.9 Å). Importantly, a decrease in the RMSD was observed for the holo L267A as compared to the apo L267A mutant (4.2 ± 0.7 and 3.9 ± 1.0 Å, respectively) as seen with the WT GAFa domain. However, no decrease in RMSD was observed for the holo F295A mutant (4.9 ± 0.9 and 4.9 ± 1.1 Å for the apo and holo F295A mutant, respectively) (Fig. 3B). Notably, RMSD measurements alone may not provide sufficient information to conclude or compare the changes in structural dynamics induced by mutating the two coevolving residue positions. Therefore, to gain deeper insights into the structural changes in the GAFa domain in the apo and holo states, we performed additional trajectory analyses, including RMSF, radius of gyration (Rg), dynamic cross-correlation (DCC), and ligand solvent-accessible surface area (SASA).

Inspecting the structural fluctuation of individual amino acid residues through root mean square fluctuation (RMSF) analysis showed that residues forming the loop regions (β1/β2, β2/β3, β4/α4, β5/β6, and β6/α5) of the domain exhibited higher fluctuations in both the apo and holo forms of the WT and mutant domains compared to other residues (Fig. 3C). Comparison of the apo and holo WT GAFa domain revealed a notable decrease in the fluctuation of the β2/β3 loop, which flanks the cGMP binding site, in the holo form. Interestingly, compared to the apo WT, a decrease in the fluctuation of this loop was also observed in the apo L267A mutant but not in the apo F295A mutant. In this regard, the holo L267A mutant exhibited a slight decrease in the fluctuation of this region compared to the apo L267A mutant. This is contrary to the behavior of the WT and F295A mutant domains, where the fluctuation of this region was notably decreased in the holo form of each domain compared to its apo form (Fig. 3C).

We subsequently performed Rg analysis of the trajectories to assess changes in the overall structure of the proteins (Fig. 3D). This revealed a decrease in the Rg of the holo WT GAFa domain in comparison to the apo (15.5 ± 0.3 and 14.6 ± 0.2 Å for the apo and holo structures, respectively) (Fig. 3D). Additionally, a comparison of the apo WT and mutant GAFa domains revealed a notable decrease in the Rg in the apo L267A mutant

compared to the apo WT, while a similar decrease was not observed for the apo F295A mutant (15.5 ± 0.3, 14.9 ± 0.2 and 15.4 ± 0.4 Å for the apo WT and the L267A and F295A mutant domains, respectively). This decrease appears similar to that observed in the holo WT domain, potentially suggesting the apo L267A mutant to be "locked" in the liganded form. However, in the holo form, the F295A mutant showed an increase in the Rg compared to holo WT, while this increase was not reflected in the holo L267A mutant (14.6 ± 0.2, 14.9 ± 0.2, and 15.3 ± 0.4 Å for the holo WT, L267A and F295A mutant domains, respectively) (Fig. 3D), potentially suggesting the holo F295A mutant to be "locked" in the apo form.

We then measured the center-of-mass distance between the two coevolving residue positions, 267 and 295, from the MD simulation trajectories to monitor their interaction and evaluate how it is altered, if at all, by mutations at these two positions (Fig. 3E). This analysis revealed a close positioning of the two residues in the apo WT GAFa domain throughout the 1000 ns-long MD simulation trajectories (Fig. 3E). A marginal increase in the distance was observed in the holo WT GAFa domain trajectories (5.3 ± 0.2 and 5.8 ± 0.9 Å for the apo and holo WT GAFa domain trajectories, respectively) (Fig. 3E). The L267A mutation resulted in an increase in the distance in the apo state, as compared to the WT GAFa domain (5.3 ± 0.2 and 6.0 ± 0.4 Å for the WT and the L267A mutant GAFa domain, respectively). The distance increased further in the holo L267A GAFa domain (6.0 ± 0.4 and 7.6 ± 1.3 Å for the apo and holo L267A mutant GAFa domain, respectively). Interestingly, the distance was found to be much larger, with greater fluctuation, in the case of the F295A mutant GAFa domain (10.9 ± 2.9 and 10.6 ± 4.9 Å for the apo and holo F295A mutant GAFa domain, respectively), in comparison to both the WT and the L267A mutant GAFa domains (Fig. 3E), suggesting a loss of the interaction between the two residues in the F295A mutant GAFa domain. Similar results were obtained upon measuring the Cα (C-alpha) distances between the residue positions.

Overall, these results indicate that the apo WT GAFa domain is more dynamic and adopts a more compact form in the holo state. Indeed, it has been suggested that cGMP binding to the GAFa domain of PDE5 converts the domain from an "open" state conformation to a more compact structure following ligand binding[22]. Similar changes have been reported for an isolated cAMP-binding GAF domain[7]. Additionally, the L267A mutant appears to shift the apo GAFa domain into a more compact structure, accompanied by a decrease in the fluctuation of the β2/β3 loop. This reduced flexibility in the apo domain, where flexibility is an important feature, could negatively impact cGMP binding. On the other hand, the F295A mutant disrupts the hydrophobic interaction between the two coevolving residue positions and other positions in the same region (Fig. 3A, E). This leads to an increase in the dynamics and a decrease in the compactness of the domain, most noticeable in the holo form, which is likely to affect the ligand binding and the subsequent cGMP-binding induced conformational change.

To investigate how these structural changes in the WT and the L267A and F295A mutant GAFa domains affect cGMP inside the ligand binding site, we determined the solvent-accessible surface area (SASA) of cGMP from the holo GAFa domain MD simulation trajectories. This analysis revealed large increases in the cGMP SASA in both the L267A (135.4 ± 143.5 Å²) and F295A (163.6 ± 150.8 Å²) mutant GAFa domains compared to the WT GAFa domain (38.8 ± 28.2 Å²) (Fig. 3F). These results suggest that the ligand is less deeply buried within the binding pocket in the mutant holo GAFa domains compared to the WT.

To understand how ligand binding may be affected by these mutations, we calculated the interaction energy (van der Waals and electrostatic) and hydrogen bonds (H-bonds) between cGMP and GAFa domain using the three independent MD simulation trajectories for each of the holo GAFa domains. This revealed a decrease in both types of interactions in the mutants (Supplementary Fig. 2), suggesting that ligand interaction with the binding site of GAFa domain is negatively affected as a result of the mutations. Overall, these results suggest that the L267A and F295A mutations lead to alterations in the structure and dynamics of the GAFa domain that are likely to affect the cGMP-binding induced conformational change.

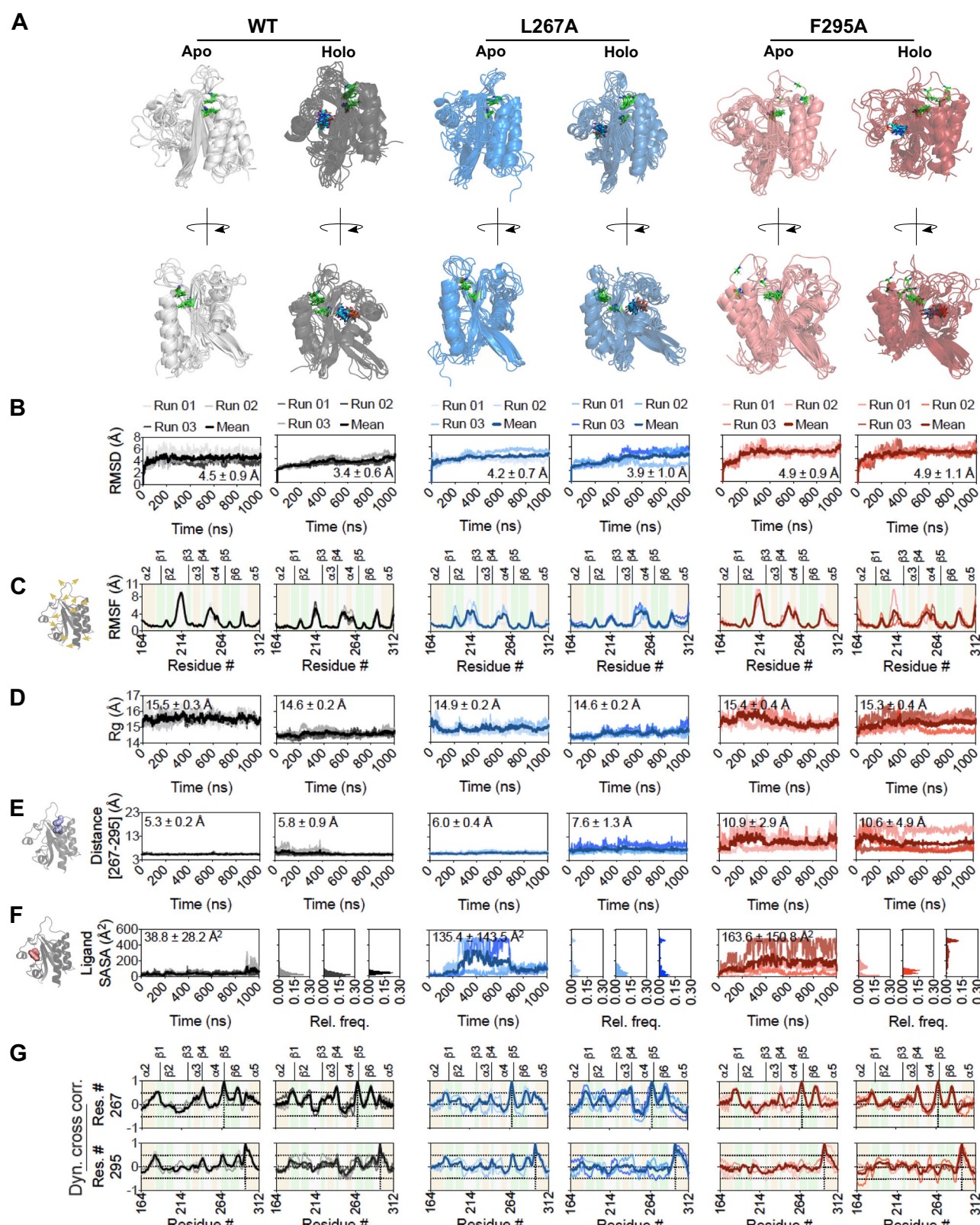

Finally, we performed dynamic cross-correlation (DCC) analysis of the MD simulation trajectories of the WT, L267A and F295A mutant GAFa domains to understand the changes in the pairwise correlated motions of residues in the proteins (Supplementary Fig. 3). A comparison of the DCC plots of the apo and holo WT GAFa domain revealed an increase in the DCC values between residue positions 244-264 (which

form the α4 region), and positions 184–210 (which form the β1/β2 region) and 264–290 (which form the β4/β5 region) (Supplementary Fig. 3). The dynamically cross-correlated motions between these residue positions were further increased in the holo L267A GAFa domain compared to the holo WT GAFa domain, while such increases were not apparent in the holo F295A mutant GAFa domain. On the other hand,

**Fig. 3 | Mutation of distant, coevolving residue positions induces dynamic structural changes in the GAFa domain of PDE5. A** Cartoon representation of GAFa domain (WT, left panel; L267A mutant, middle panel; and F295A mutant, right panel) dynamics across the 1000 ns-long MD simulation runs captured using a composite image of snapshots that are 200 ns apart. The ligand is shown as blue carbon sticks, and positions 267 and 295 are shown as green carbon sticks in each complex. **B** Graphs showing root mean squared deviation (RMSD) measurements of the apo and holo GAFa domains (WT and mutants) across the 1000 ns-long MD simulation runs. **C** Schematic (left panel) and graphs showing root mean square fluctuation (RMSF) measurements of the apo and holo GAFa domains (WT and mutants) across the 1000 ns-long MD simulation runs (right panel). **D** Graphs showing radius of gyration (Rg) values of the apo and holo GAFa domains (WT and mutants) across the 1000 ns-long MD simulation runs. **E** Schematic highlighting residues L267 and F295 (left panel) and graphs showing center-of-mass distance between amino acid positions 267 and 295 in the apo and holo GAFa domains (WT and mutants; right panel). Note how the hydrophobic interaction is disrupted in the F295A mutant domain as is evident from the increased distance between the two positions, as opposed to the WT and L267A mutant GAFa domains that maintained the distance across the 1000 ns-long MD simulation runs. **F** Schematic highlighting ligand (cGMP) surface (left panel) and graph showing solvent-accessible surface area (SASA) of cGMP across the 1000 ns-long MD simulation runs of the holo GAFa domain (WT and mutants; right panel). Outsets, frequency distribution of ligand SASA across the 1000 ns-long MD simulation runs of each complex. **G** Graphs showing dynamic cross-correlation (DCC) values of residue positions 267 (top panel) and 295 (bottom panel) in the apo and holo GAFa domain (WT and mutants) obtained from the 1000 ns-long MD simulation runs. Vertical dotted lines indicate residue 267 in graphs in the top panel and residue 295 in graphs in the bottom panel, while horizontal dotted lines indicate DCC values of 0.5 and −0.5.

the apo L267A mutant GAFa domain appears to disrupt the dynamic cross-correlated motions that were originally observed in the apo WT GAFa domain. In contrast, the F295A mutant GAFa domain did not show any notable impact on these motions (Supplementary Fig. 3).

We then focused our attention on the DCC of the residue positions 267 and 295 against all other residues in the domain (Fig. 3G). While both positions showed highly correlated motions (equal to or above 0.5) with residues in the α2/β1 loop, sheets β4 (which flanks the cGMP binding site) and β6 in the apo WT GAFa domain trajectories, position 295 showed additional correlated motions with residues on the C-terminal side in the helix α5 (Fig. 3G). The residue position 267 showed increased correlated motions with residues in the β2/β3 loop, while residue position 295 showed a decrease in correlated motions with residues in the α2/β1 loop in the holo, as compared to the apo, WT GAFa domain. A decrease in the correlated motions of residue positions 267 and 295 with residues in sheets β4, which contains the residue at the 267 position, was observed in the apo L267A and F295A mutant, as compared to the WT GAFa domain. It is possible that disruption in the inter-residue interaction caused by mutating the 267 and 295 positions results in the observed decrease in DCC motions between these positions and sheet β4 structure, which may have an effect on ligand binding. Further, no apparently large differences were observed for the correlated motions of residue positions 267 and 295 in both the holo L267A and F295A mutant, as compared to the WT, GAFa domain (Fig. 3G). Overall, these results indicate a decrease in the dynamically cross-correlated motions between the two coevolving positions and specific regions in the L267A and F295A mutants compared to the WT GAFa domain.

### L267A and F295A mutations induce a conformational change and result in an increase in the $EC_{50}$ of cGMP-binding induced conformational change in PDE5 GAFa domain

Having observed alterations in the structural dynamics in the GAFa domain of PDE5 upon L267A and F295A mutations using MD simulations, we then proceeded to determine the impact of these mutations experimentally. In this regard, structural studies of the PDE5 GAFa domain have revealed the conformational changes that occur in the protein upon cGMP binding[16,22] (Fig. 4A). Broadly, these changes include a close juxtaposition of the sheet β3 and helix α4 leading to a "closing" of the cGMP binding site, an "inward" movement of sheet β1 and β2 towards the cGMP binding site and an "outward" movement of the loop β2/β3 (Fig. 4A, left panel). Additionally, cGMP binding is associated with a close juxtapositioning of helices α2 and α5 at the N- and C-termini of the GAFa domain, likely associated with the transduction of the allosteric signal from the GAFa domain to the catalytic domain in PDE5 (Fig. 4A, right panel). We decided to take advantage of the latter to monitor cGMP-binding induced conformational change in the GAFa domain through an increase in the BRET efficiency of the GAFa domain biosensor that we have described previously[24] (Fig. 4B). BRET is a biophysical technique that involves resonance energy transfer between a bioluminescent donor and a fluorescent acceptor, the extent of which is dependent on the spectral overlap, distance, and relative orientation between the two reporters. In this case, the biosensor consists of an isolated GAFa domain of PDE5 sandwiched between $GFP^2$ at the N-terminal side (BRET acceptor) and RLuc at the C-terminal side (BRET donor). $GFP^2$ is a brighter variant of GFP with a F64L mutation and has an emission spectra similar to that of GFP but with significantly blue-shifted excitation spectra with a peak of 396 nm (Patent no. US 6.12,188 B1)[29], leading to a substantial spectral overlap of RLuc emission with $GFP^2$ excitation as well as a large separation of RLuc and $GFP^2$ emissions[24–26]. Since the two protein reporters are the same for the WT and mutant sensors, then any change in the BRET signal would suggest a change in the distance and/or orientation of the protein reporters, which can only happen if there is a conformational change in the domain as a result of mutation (change in basal BRET) or ligand-binding (change in cGMP-binding induced BRET).

We generated L267A and F295A mutations individually in the BRET-based GAFa domain biosensor plasmid[24] (Fig. 4C) and confirmed the expression of the biosensors through Western blot analysis (Fig. 4D, Supplementary Fig. 4). Determination of bioluminescence spectra revealed a second peak in the WT GAFa domain biosensor that was not observed in the RLuc-alone control (Fig. 4E), indicating a significant resonance energy transfer from RLuc to $GFP^2$. On the other hand, the mutant GAFa domain biosensors showed a reduced second peak, indicating a decrease in energy transfer from RLuc to $GFP^2$. Further, basal BRET ratio measurements showed a statistically significant decrease in the mutants ($0.76 \pm 0.05$, $0.25 \pm 0.03$, and $0.26 \pm 0.02$ for WT, L267A, and F295A, respectively; $p < 10^{-4}$ for WT vs L267A and $p < 10^{-4}$ for WT vs F295A), suggesting a conformational change in the L267A and F295A mutant GAFa domains.

Further, we incubated the biosensors with 1 µM cGMP, a relatively high concentration as the PDE5 GAFa domain binds cGMP with nano-molar affinity[24], and measured BRET. In agreement with our previous report[24], the WT GAFa domain biosensor showed an increase in BRET in the presence of 1 µM cGMP (Fig. 4F), reflecting a change in the conformation of the protein. On the other hand, neither the L267A nor the F295A mutant biosensors showed a discernible change in BRET in the presence of 1 µM cGMP (Fig. 4F). These results indicate that the mutations resulted in either a decrease in the affinity of the GAFa domain for cGMP or alteration in the structure of the domain in a way that cGMP binding does not induce a conformational change that could be detected using BRET. To determine which of these possibilities is responsible for the lack of an increase in BRET, we performed dose-response experiments with a range of cGMP concentrations. This revealed a cGMP concentration-dependent increase in BRET of WT GAFa domain biosensor, with an $EC_{50}$ value of $19 \pm 13$ nM (Fig. 4G, H), which is similar to the previously reported value[24]. Interestingly, both the L267A and F295A mutant GAFa domain biosensors showed increase in BRET similar to that of the WT protein, albeit at much higher concentrations of cGMP with apparent $EC_{50}$ values of ~110 ± 74 and ~194 ± 71 µM for the L267A and the F295A mutant, respectively (~5800- and 10,000-fold higher for the L267A and the F295A mutants than the WT protein; with $p = 10^{-4}$ for WT vs L267A and $p < 10^{-4}$ for WT vs F295A; the actual $EC_{50}$ values are likely to be higher as the BRET increases did not saturate at the highest concentration of cGMP tested here) (Fig. 4G, H). Together, these results suggest that the L267A and F295A mutations,

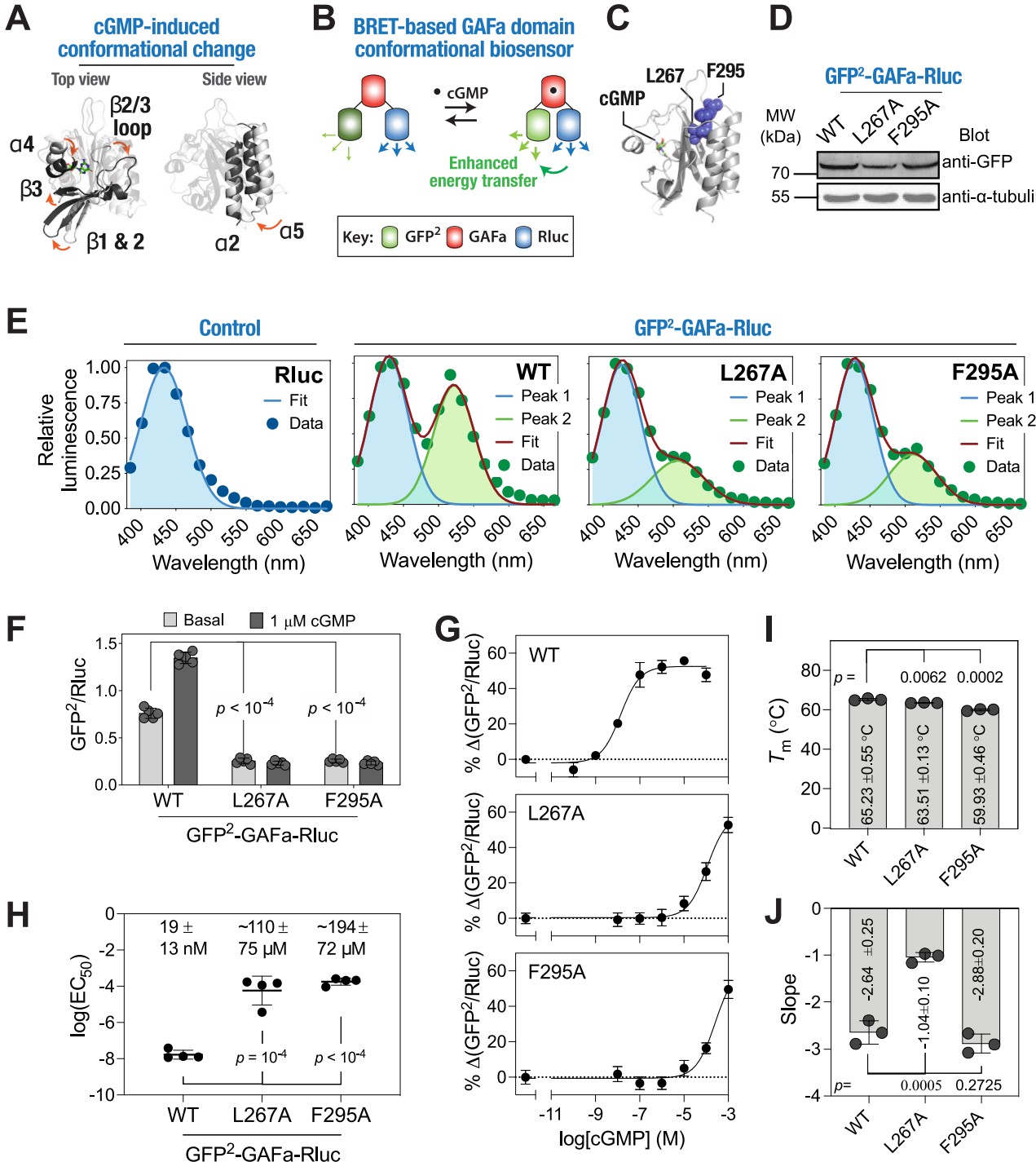

although positioned away from the cGMP binding site, decreased the cGMP affinity of the GAFa domain.

Given that the L267A and F295A mutants showed a reduction in the basal BRET, we attempted to determine the impact of these mutations on the thermal stability of the GAFa domain. We posited that thermal denaturation of the GAFa domain would lead to a decrease in BRET of the biosensor constructs[30]. For this, we utilized the thermally stable fluorescent protein, mNeonGreen, as the BRET acceptor and the NLuc luciferase protein as the BRET donor, instead of GFP[2] and RLuc due to their low thermal stability[31]. We generated plasmid expressing mNG-GAFa-NLuc biosensors, either WT or L267A or F295A mutant, and prepared lysates after transfecting HEK293T cells. Lysates were then incubated at a range of

temperatures (ranging from 30 to 80 °C) for 10 min, following which BRET was measured. BRET data were fitted to a Boltzmann sigmoidal model to determine the $T_m$, the temperature at which the relative BRET ratio is half of its maximum value (Supplementary Fig. 5). This revealed $T_m$ values of 65.23 ± 0.55, 63.51 ± 0.13, and 59.93 ± 0.46 °C for the WT, L267A and F295A mutant GAFa domains ($p = 0.0062$ and 0.0002 for WT vs L267A and WT vs F295A, respectively) (Fig. 4I), suggesting that the mutations resulted in a minor, although statistically significant, decrease in the thermal stability of the GAFa domain. Importantly, the temperature-dependent decrease in BRET of the L267A was found to be steeper in comparison to the WT and F295A mutant GAFa domains (slope of −2.64 ± 0.25, −1.04 ± 0.10 and −2.88 ± 0.20 for WT, L267A and F295A mutant, respectively; $p = 0.0005$ for

**Fig. 4 | Mutation of the distant, coevolving residues alters cGMP-induced allosteric conformational change in the isolated GAFa domain of PDE5.**
**A** Cartoon representation of apo (PDB: 3MF0[16]) and holo (PBD: 2K31[22]) PDE5 GAFa domain structures aligned using all Cα-atoms (left panel) and using Cα-atoms of α2 helix (right panel) highlighting structural changes (apo: gray, holo: black) induced upon cGMP binding. **B** Schematic showing the BRET[2]-based GAFa domain conformational biosensor that shows increased energy transfer upon cGMP binding likely due to the structural juxtaposition of helix α5 against helix α2, and thus, bringing the BRET donor and acceptor closer. **C** Cartoon representation of the PDE5 GAFa domain highlighting the distant, coevolving residues L267 and F295 (spheres in blue) that were mutated to A. The ligand, cGMP, is shown in the stick representation. **D** Western blot analysis of the cell lysates prepared from HEK293T cells transfected with either the WT or mutant GAFa domain biosensor plasmids and probed using an anti-GFP antibody showing the expression of biosensor constructs. Whole blot image used for generating the figure is shown in Supplementary Fig. 4. **E** Graphs showing bioluminescence spectra obtained from lysates prepared from cells expressing either RLuc alone (control) or the indicated GAFa biosensor constructs (containing the WT or the mutant PDE5 GAFa domains sandwiched between GFP[2] and RLuc proteins). Note the appearance of a second peak in the WT GAFa domain biosensor, indicating energy transfer from RLuc (donor) to GFP[2] (acceptor), while this peak is reduced in both L267A and F295A mutant GAFa domain biosensors. Data shown are representative of multiple experiments and were fitted using either a single (RLuc alone) or two Gaussian (GAFa domain biosensors)

distributions. **F** Graph showing BRET values (ratio of GFP[2] and RLuc emissions) of the WT, L267A, and F295A mutant GAFa domain biosensors in the absence and presence of cGMP (1 µM). Note the significant reduction in the basal BRET values of the L267A and F295A mutant biosensors compared to the WT biosensor. Data shown are the mean ± standard deviation (S.D.; error bars) from five measurements of a representative experiment, with experiments performed four times. **G** Graphs showing percentage increase in BRET values for the WT, L267A, and F295A mutant GAFa domain biosensors with the indicated cGMP concentrations. Note the shift in the cGMP dose-response curves of the L267A and F295A mutant GAFa domain biosensors in comparison to the WT GAFa domain. Data shown are the mean ± standard deviation (SEM.; error bars) from a representative experiment, with experiments performed four times. **H** Graph showing log($EC_{50}$) values for cGMP-induced conformational change in the WT, L267A, and F295A mutant GAFa domain biosensor. Data shown are mean ± S.D. obtained from four independent experiments. Inset, values on top indicate the cGMP $EC_{50}$ values (mean ± S.D.) of the respective GAFa domain biosensors. (**I, J**) Graph showing melting temperatures ($T_m$) and slope of the melting temperature curves of the WT and L267A and F295A mutant GAFa domain biosensors. Data shown are mean ± S.D. obtained from three independent melting temperature curves (curves are provided in Supplementary Fig. 9). All $p$-values shown in the figure were obtained from Student's $t$-tests (two-tailed, unpaired, equal variance) performed for the L267A or F295A mutants against the WT GAFa domain. For all tests, a $p$-value of <0.05 was considered statistically significant.

WT vs L267A mutant) (Fig. 4J). These results suggest that the L267A mutation leads to a resistance to thermal denaturation in the initial phases (below 60 °C), in a broad agreement with the decreased Rg, and thus, the structural compactness, observed in MD simulations of the apo L267A mutant (Fig. 3D). Overall, these results suggest that the lower basal BRET and higher apparent $EC_{50}$ of cGMP-binding induced conformational change observed in the L267A and F295 A mutants are likely not due to a decrease in the thermal stability of the proteins but instead reflect changes in their conformation and structural dynamics.

## L267A and F295A mutations induce a conformational change and significantly reduce the cGMP-binding induced conformational change in the full-length PDE5A2

Having elucidated the impact of the L267A and F295A mutations in the isolated GAFa domain, we aimed to investigate their effect on the cGMP-binding induced conformational change in the full-length PDE5 protein. For this, we incorporated the mutations in our previously described BRET-based, full-length PDE5A2 biosensor (Fig. 5A) that reports a conformational change upon cGMP binding to the GAFa, but not the catalytic, domain in the protein[25]. Western blot analysis revealed similar expression of the WT and mutant biosensor constructs (Fig. 5B; Supplementary Fig. 6). Bioluminescence spectral analysis revealed a second peak corresponding to GFP[2] suggesting resonance energy transfer from RLuc in the case of the WT PDE5A2 biosensor (Fig. 5C). Importantly, as seen with the isolated GAFa domain biosensor, a decrease in the resonance energy transfer was observed for L267A and F295A mutant biosensors (Fig. 5C), leading to a significant decrease in basal BRET, as compared to the WT biosensor (Fig. 5D) (0.222 ± 0.009, 0.115 ± 0.002, and 0.122 ± 0.004 for WT, L267A, and F295A, respectively, $p < 10^{-4}$ for both WT vs L267A and WT vs F295A), suggestive of a conformational change in the mutant proteins. Further, dose-response experiments with a range of cGMP concentrations revealed both a decrease in the maximum BRET change (-26.60 ± 3.42, −17.43 ± 1.31, and −12.60 ± 2.37% for WT, L267A, and F295A, respectively) and a rightward shift in the dose-response curve in the mutant biosensors, with $EC_{50}$ values of 73 ± 20 nM, 181 ± 30 nM, and 3.4 ± 2.1 µM for WT, L267A, and F295A, respectively, $p = 0.0071$ and 0.0496 for WT vs L267A and WT vs F295A, respectively (Fig. 5E). These results indicate a change in the conformation in the absence of cGMP and a substantial decrease in cGMP affinity of the GAFa domain in the L267A and F295A mutant full-length PDE5A2 biosensors. Interestingly, the increase in the $EC_{50}$ was greater in the F295A mutant compared to the L267A mutant indicating a more detrimental effect seen when mutating the F295 position in full-length PDE5A2.

Having determined the $EC_{50}$ values of cGMP-binding induced conformational change in both the isolated GAFa domain and the full-length PDE5A2, we compared these values with the free energy changes ($ΔG$) of cGMP binding to the WT and mutant GAFa domains as determined from the MD simulation trajectories. This revealed a general decrease in the free energy changes of cGMP binding in both L267A and F295A mutant GAFa domains, although the effect was less pronounced in the case of the L267A mutant (Fig. 5E,F). Further, a clear difference could be observed in the impact of the two mutations in the isolated GAFa domain and the full-length PDE5A2 biosensors in terms of $EC_{50}$ of cGMP-binding induced conformational change (Fig. 5F), suggesting that additional domains such as the GAFb and the catalytic domain in the full-length proteins (Fig. 5A) have a role in the cGMP-induced conformational change in the GAFa domain.

To further support the important role of the two coevolving residues in GAF domains, a multiple sequence alignment of human cyclic nucleotide-binding PDEs revealed that these two amino acids are conserved in both the GAFa and GAFb domains of these PDEs (Supplementary Text 1). Additionally, analysis of variants at these positions in the PDE5A2 gene available in the Genome Aggregation Database (gnomAD) revealed a potentially harmful impact of the variations, further supporting their critical role in the functioning of the associated PDE5A2 (Supplementary Table 2) (See Supplementary Methods and Supplementary Results). Notably, the N-terminal loop of PDE5 has been suggested to play a role in the cGMP-induced conformational allostery[16]. Therefore, the choice of PDE5 isoform could have important implications for understanding the allosteric regulation of PDE5 catalytic activity, as the PDE5A2 isoform, which we utilized in the design of the full-length PDE5 biosensor, is 42 amino acids shorter than the PDE5A1 isoform.

## Mutation of the distant coevolving residues in the fluorescent GAF domain protein, miRFP670nano3, decreases its fluorescence

Having established the impact of the distant coevolving residue positions 267 and 295 in the GAFa domain of PDE5, we aimed to determine the general applicability of these findings with respect to ligand binding in GAF domains in general. For this, we analyzed the structures of a variety of GAF domains available in the RSCB database and found the two residue positions to be in close proximity to each other and distant from the bound ligand (in cases where the structures contained a ligand), suggesting a conserved role for the two residues (Supplementary Fig. 7).

To confirm the role of the two coevolving residues in GAF domain function, we utilized a recently reported GAF domain fluorescent protein,

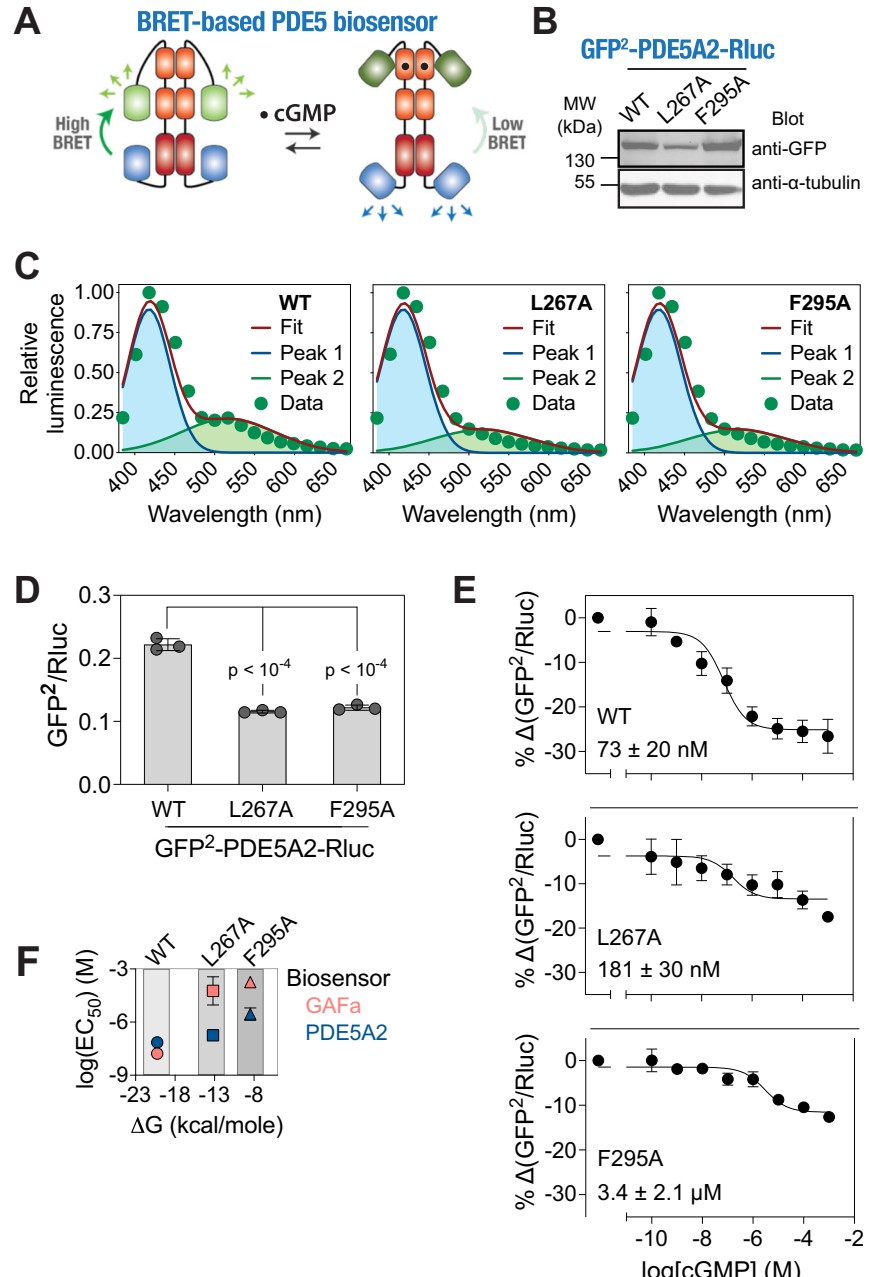

**Fig. 5 | Mutation of distant, coevolving residues alters cGMP-induced allosteric regulation in the full-length PDE5.** **A** Schematic showing the mechanism of the BRET[2]-based, full-length PDE5A2 conformational biosensor in the absence and presence of cGMP. **B** Western blot analysis of the cell lysates prepared from HEK293T cells transfected with either the WT or mutant full-length PDE5A2 biosensor plasmid constructs and probed using an anti-GFP antibody showing the expression of biosensor constructs. Whole blot image used for generating the figure is shown in Supplementary Fig. 6. **C** Graphs showing bioluminescence spectra obtained from lysates prepared from cells expressing the WT and L267A and F295A mutant PDE5 biosensor constructs. Note the reduction in the GFP[2] emission peaks in the L267A and F295A mutant PDE5 biosensors. Data shown are mean ± S.D. from a representative experiment, with experiments performed three times. **D** Bar graph showing the basal BRET values of the WT, L267A, and F295A mutant PDE5 biosensors. Note the significant reduction in the basal BRET values of the L267A and F295A mutants compared to the WT GAFa domain. **E** Graphs showing percentage

decrease in BRET values of the WT, L267A, and F295A mutant PDE5 biosensors after 30 min incubation with the indicated cGMP concentrations. Note the decrease in the maximum % change in BRET as well as the rightward shift in the cGMP dose-response curves of the mutant biosensors. Inset, values indicating the $EC_{50}$ of cGMP-induced conformational change for the respective GAFa domains in the full-length PDE5 biosensor. For (**D**, **E**), data shown are mean ± S.D. from three experiments, with each experiment performed in triplicates. **F** Graph showing $\log(EC_{50})$ (mean ± S.D.) values of cGMP-binding induced conformational change in the isolated GAFa domain and full-length PDE5 BRET-based biosensors against ΔG (mean ± S.D.) values obtained from the MD simulation runs of the isolated GAFa domain (WT and mutants). Data shown are mean ± S.D. from three experiments. All *p*-values shown in the figure were obtained from Student's *t*-tests (two-tailed, unpaired, equal variance) performed for the L267A or F295A mutants against the WT GAFa domain. For all tests, a *p*-value of <0.05 was considered statistically significant.

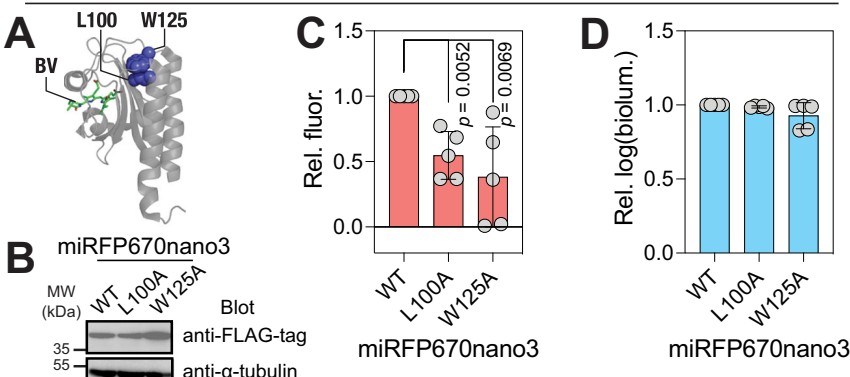

**Fig. 6 | Mutating the distant coevolving residue positions affects the function of other ligand-binding GAF domains. A** Cartoon representation of the miRFP670nano3 GAF domain (PDB: 7LSC) highlighting distant coevolving residues equivalent to PDE5 L267 and F295 (L100 and W125, respectively) and the ligand, biliverdin (BV). **B** Western blot analysis of the cell lysates prepared from HEK293T cells transfected with either the WT or mutant miRFP670nano3 plasmid constructs and probed using an anti-FLAG antibody showing the expression of

biosensor constructs. Whole blot image used for generating the figure is shown in Supplementary Fig. 8. Graphs showing relative fluorescence (**C**) and bioluminescence (**D**) of the WT, L100A and W125A mutant miRFP670nano3. Data shown are mean ± S.D. obtained from five independent experiments. The *p*-values shown in (**C**) were obtained from Student's *t*-test (two-tailed, unpaired, equal variance). For all tests, a *p*-value of <0.05 was considered statistically significant.

miRFP60nano3[12,32,33]. It is a small, single-domain, engineered near-infrared fluorescent protein derived from cyanobacteriochrome photoreceptor GAF domain (Fig. 6A) that covalently binds the chromophore, biliverdin[12,32,33]. We expressed the WT and L100A and W125A mutants (equivalent to the L267A and F295A GAFa domain mutations, respectively; based on structural alignment) (Fig. 6A), in HEK293T cells. We included a C-terminal fusion of a luciferase protein, picALuc(E50A)[34], for monitoring protein expression levels. Protein expression was also assessed using Western blot assay, which showed similar expression of the proteins (Fig. 6B; Supplementary Fig. 8). Fluorescence measurements revealed a significantly reduced values in cells expressing the L100A and W125A mutant miRFP670nano3, as compared to the WT protein (relative fluorescence of 1.00 ± 0.00, 0.55 ± 0.18, 0.38 ± 0.38 for WT, L100A, and W125A, *p* = 0.0052, 0.0069 for WT vs L100A and WT vs W125A, respectively) (Fig. 6C) while the expression of the proteins was found to be similar (Fig. 6B,D). These results indicate that the distant, coevolving residues are likely required for the miRFP670nano3 GAF domain to bind its ligand, biliverdin, required for its fluorescence activity and thus confirm that the two coevolving residues are important for GAF domain function.

## Elucidating cGMP-induced allosteric activation of PDE5 GAFa domain

Having established the impact of mutations in the distant, coevolving residues on GAF domain function in the PDE5 GAFa domain and miRFP670nano3 proteins, we aimed to better understand the mechanism of cGMP-binding induced conformational change and the impact of these mutations on this mechanism. This was achieved through the analysis of MD simulation trajectories of the apo and holo GAFa, WT and mutant, domains. Specifically, we measured the distance between sheet β3 and helix α4 surrounding the cGMP binding site (Fig. 7A, left panel), as the two secondary structural elements act as a "lid" that gates the cGMP binding site and have been reported to "close in" upon cGMP binding (cGMP-binding signal)[22]. Additionally, we measured the distance between helices α2 and α5 that are located at the N- and C-termini of the domain, respectively. These helices have been found to move closer upon cGMP binding, which likely represents the allosteric activation signal (Fig. 7A, right panel). Following measurement of the two distances—i.e. the cGMP binding-associated distance (β3-α4 distance) and allosteric activation-associated distance (α2-α5 distance)—we pooled the distances from all three 1000 ns-long MD simulation trajectories and plotted the probability densities of the distances

for the apo and holo GAFa, WT and L267A and F295A mutant, domains (Fig. 7B). This analysis revealed that the β3-α4 and α2-α5 distances in the apo, WT GAFa domain largely centered around 11 and 13 Å, respectively, while the distances in the holo, WT GAFa domain centered around 10 and 9 Å, respectively, (Fig. 7B), indicating that cGMP binding to the WT GAFa domain results in a decrease in both the β3-α4 and α2-α5 distances, i.e. a "closing" of the cGMP binding site "lid" and a "closing in" of the N- and C-terminal α-helices. The latter is likely the basis for the increase in BRET in the presence of cGMP that we observed with the WT GAFa domain biosensor (Fig. 4F,G). Similar findings were reported for the isolated GAF domain of Anabaena adenylyl cyclase utilizing amide hydrogen/deuterium exchange mass spectrometry[7].

In the apo L267A mutant GAFa domain, the β3-α4 and α2-α5 distance densities were centered similar to those in the WT GAFa domain (Fig. 7B). In the holo L267A mutant, the α2-α5 distance density center was decreased (from 13 to 9.5 Å) to levels similar to that of the WT GAFa domain (9 Å). However, the β3-α4 distance density center was found to increase (from 11.5 to 14 Å), rather than decrease as seen with the WT GAFa domain (Fig. 7B). This suggests that the L267A mutation results in the opening of the "lid" of the cGMP binding site and thus, likely, enhancing the unbinding of cGMP from the protein leading to an increase in the $EC_{50}$ of cGMP-binding.

On the other hand, in the apo F295A mutant GAFa domain, two separate density centers (one with β3-α4 and α2-α5 distances of 10 and 11.5 Å, respectively, and another with distances of 13.4 and 13 Å, respectively) were observed (Fig. 7B), suggesting a destabilization effect of the mutation in the apo state affecting both the "lid" of the cGMP binding site and the terminal helices. However, in the holo state, in which the WT GAFa domain assumed a more compact form in comparison to the apo form, the F295A mutant displayed a density center that is similar to the WT holo state with regards to the β3-α4 distance (10 and 10 Å for the F295A and WT GAF domains, respectively), but the α2-α5 distance found to be similar to that of the major center seen in the apo state of the F295A mutant protein (11.5 and 11.5 Å, respectively) with a loss of the second center observed in the apo state, which was higher than the distance observed with the holo WT GAFa domain (11.5 and 9.5 Å, respectively) (Fig. 7B). These results suggest that cGMP binding to the F295A mutant may not lead to the effective transduction of the allosteric signal, which agrees with an increase in the $EC_{50}$ of cGMP-binding induced conformational change determined using the GAFa biosensor (Fig. 4F, G). Overall, these results illustrate the impact of L267A and F295A mutations on the conformational landscape of the GAFa

**Fig. 7 | Altered ligand binding site and allosteric activation upon mutation of distant, coevolving residues in the PDE5 GAFa domain. A** Cartoon representation of the GAFa domain highlighting helix α4 and sheet β3 from the ligand binding site that "closes" upon cGMP binding (left panel) and N-terminal helix α2 and C-terminal helix α5 that move closer upon cGMP binding (right panel). **B** Graphs showing probability distribution (kernel density estimation) of distances between helix α4 and sheet β3 (x-axis) and N-terminal helix α2 and C-terminal helix α5 (y-axis) of the apo (upper panel) and holo (lower panel) GAFa domain (WT and mutants) determined from a cumulative of three independent, 1000 ns-long MD simulation trajectories. Color bars, probability densities.

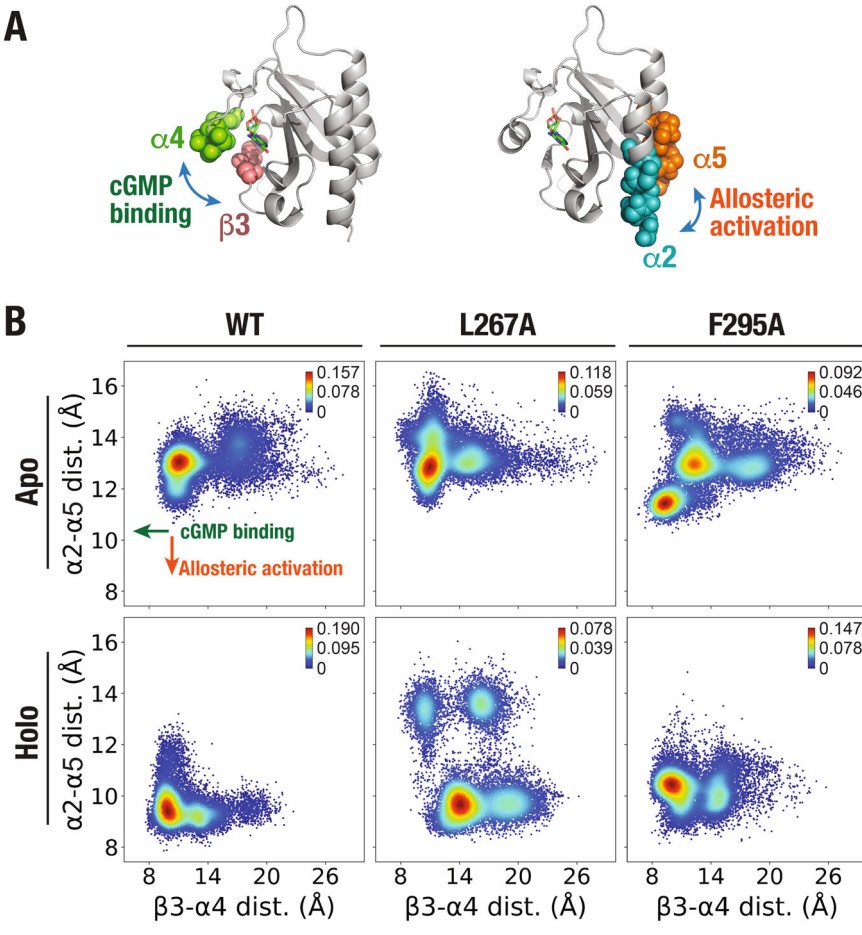

domain and point to a critical role played by the two residues in the cGMP-binding induced conformational change in the GAFa domain, likely through two distinct mechanisms.

## Discussion

GAF domains are among the most conserved molecular switches that are abundantly present in different proteins across a diverse range of species. In this study, we used SCA to identify a cluster of highly coevolving residue positions in GAF domains. The SCA method has been successfully applied in a number of cases, including decoding the functional evolution of enzymes[35] and their metal binding specificities[36], discrimination between correct and incorrect protein folds that are based on de-novo structure prediction[37], and predicting allosteric networks in proteins[38–42] as well as functional RNA molecules[43]. We identified two residue positions with high coevolutionary scores that are located distant from the ligand binding site. Mapping the two residue positions onto GAF domain structures from different species revealed that the two residues are juxtaposed and may play a role in the allosteric function of the domain. We used the GAFa domain of PDE5 to investigate the role of the two residue positions (L267 and F295) in the domain. Our in-silico analysis of the apo and holo forms of the domain, as well as in vitro assays, shed some light on the functional role of the two positions on the ligand-binding induced conformational change in the domain. Mutating the L276 residue, which is located in the sheet β5, induces a structural change in the domain that we detected experimentally as a decrease in BRET. This change in the conformation is associated with a decrease in the structural dynamics of the domain and shifts the apo state into a more compact form, as can be inferred from the RMSD, RMSF, and Rg analyses of the MD trajectories. This, in turn, could potentially affect ligand binding, as the structural flexibility of the apo form is an important feature that facilitates ligand binding. To further support the idea of these

structural alterations, our in vitro assays showed that the L267A mutant is more resistant to the thermal unfolding with increasing temperature, although the $T_m$ value was slightly decreased compared to the WT. Additionally, our MD simulation analysis showed that, in the holo form, the L267A mutation results in the opening of the α3-β4 "lid" that gate the cGMP binding site, making it difficult to retain the ligand within the binding site. This was reflected by an increase in ligand SASA and a decrease in inter-action energy, hydrogen bond formation, and binding free energy in the L267A mutant domain compared to the WT. These computational findings were further confirmed experimentally using an L267A mutant BRET-based GAFa domain biosensor, which showed a more than 5000-fold increase in the $EC_{50}$ of the cGMP-binding induced conformational change compared to the WT GAFa domain.

On the other hand, the F295 residue is located on the β6/α5 loop near the terminal helix (α5), which transmits the allosteric signal to the catalytic domain of the protein. Our results show that the F295A mutation induces a structural change in the domain, which was experimentally detected as a decrease in the basal BRET ratio of the F295A mutant GAFa-domain biosensor. Based on the MD simulation trajectory analysis, this mutation disrupts the inter-residue interactions originally formed between this position and neighboring residues, including the L267 residue, in the WT GAFa domain. This results in an increase in the structural dynamics of the domain, as evidenced by RMSD analysis, and in a less compact structure in the holo form, as indicated by Rg analysis. Specifically, the F295A mutation increases the α2-α5 distance in the holo form, suggesting that it interferes with the allosteric signal transduction to the catalytic domain following ligand binding. These findings were further validated experimentally using an F295A mutant BRET-based GAFa-domain biosensor, which showed more than 10000-fold increase in the $EC_{50}$ of the cGMP-binding induced con-formational change compared to the WT GAFa domain.

Overall, these results indicate that the two mutations negatively impact cGMP-binding induced conformation change in the isolated GAFa domain. The L267A mutation appears to directly affect ligand binding by decreasing the flexibility and increasing the rigidity of the apo form, both of which are important for ligand binding in the apo form. Additionally, it apparently results in the opening of the α3-β4 "lid", which gates the cGMP binding site, making it difficult to retain the ligand inside the binding pocket in the holo form. On the other hand, the F295A mutation appears to negatively affect cGMP-binding induced conformational change by two mechanisms, both as a result of disrupting the local inter-residue hydrophobic interaction. First, this leads to the destabilization of the ligand inside the binding pocket, which directly and negatively affects ligand binding. Second, this disruption of the hydrophobic interaction resulting from mutating this position leads to a change in the cGMP allosteric signal transmission to the terminal α5 helix, which extends to connect GAFa to GAFb domain in PDE5 and transmits the allosteric signal to the catalytic domain through the GAFb domain.

Furthermore, the structural changes induced by the two mutations in the isolated GAF domain were also recapitulated experimentally in the full-length enzyme as a decrease in the basal BRET ratio of the mutant PDE5-based biosensors. Moreover, the mutant biosensors displayed a suboptimal conformational change as indicated by the lower values of the maximal change in BRET ratio at the highest cGMP concentration tested. Additionally, there was a significant decrease in the ligand-binding induced conformational change in the mutant proteins, as indicated by the rightward shift of the cGMP dose-response curve and the increase in the $EC_{50}$ values of cGMP-binding induced conformational change. This indicates that the two positions play an important role in the allosteric function of the GAFa domain in full-length PDE5. Importantly, the F295A full-length PDE5 mutant showed a greater decrease in the $EC_{50}$, further supporting the results reported here that the F295A mutant may also affect the allosteric signal transduction from GAFa domain to catalytic domain following ligand binding. It is important to note that the ligand-induced conformational change initiated in the GAFa domain is transmitted to the catalytic domain of PDE5 through the GAFb domain. Consequently, this allosteric communication is expected to be altered in the absence of the GAFb domain. While this is true, our focus was to investigate how changes in ligand-induced conformational allostery observed in the isolated, mutant GAFa domains translate to the full-length PDE5 as an endpoint. Furthermore, we show that the functional role of the two positions is not restricted to the GAFa domain of PDE5 but extends to other GAF domains. For instance, mutating the two positions in the engineered GAF domain fluorescent protein, miRFP670nano3, resulted in a decline in its fluorescent emission, likely due to reduced biliverdin binding.

In this regard, the near full-length structure of homodimeric PDE2A, which includes both the regulatory and catalytic domains, provided some insights into GAF domain-mediated regulation of PDE catalytic domain[44]. In the absence of cGMP binding, the two catalytic domains in the dimer occlude the substrate binding site in each other by means of a loop region (H-loop). Binding of the allosteric ligand, cGMP, to the GAFb domain causes the H-loop to "swing" out of the catalytic pocket, allowing access to the catalytic site for the substrate, cGMP, molecules[44]. Relevant to this, a previous structural study of PDE6 tandem GAF domains (GAFa and GAFb), which shows the highest sequence and structural homology with tandem GAF of PDE5, reported cGMP-dependent structural changes in the GAFa domain that are similar to the GAFa of PDE5, including movement of the β1/β2 loop, the α2/α3 region, and α4 helix[45]. However, the same study did not report a cGMP-dependent change in the conformation and distance between the terminal helices, α2 and α5, which is the basis for the design of our in-house PDE5 GAFa-domain biosensor[24]. Additionally, the study suggested that allosteric signal transmission from the GAFa domain to the GAFb domain takes place at several sites where the two domains meet. This includes interactions from the GAFa β1/β2 loop to the GAFb β4/β5 and β6/α5 loops, as well as from the GAFa α2/3 region to the GAFb β6/α5 loop[45]. In fact, some of the reported PDE6 disease-causing mutations are located in the vicinity of the β1/β2 loop of GAFa[46–49],

suggesting a disruption in the allosteric communication caused by these mutations in PDE6. On the other hand, our computational and biophysical investigations of the GAF domain provide evidence of additional, coevolving amino acid residues that are involved in the cGMP-binding induced conformational change. These coevolving residues are not only located distant from the cGMP binding site but also do not show interfacial interaction with the GAFb domain, suggesting a more complex and multimodal allosteric signal relay mechanism across cyclic nucleotide-binding PDEs than what has been proposed in previous studies. We note that the current work did not intend to examine the effect of different force fields, such as AMBER FF, on the biomolecular simulation systems, which could be explored in future work.

Moreover, our results show that the two coevolving residues are invariant across the GAFa and GAFb domains of all human cyclic-nucleotide binding PDEs, and mutations at these two positions in human PDEs, reported in the gnomAD database, were predicted to be potentially harmful to the associated PDE genes. This strongly suggests a critical role for these two residues in the GAF domain function, specifically in ligand binding and induction of allostery in PDEs. On the other hand, the F295 position is part of the NKFDE motif (N286, K287, F295, D299, E230, numbering according to human PDE5A1), which distinguishes cyclic nucleotide-binding GAF domains from other GAF domains. Since they are conserved in cyclic nucleotide binding GAFs, the residues making up this motif were initially thought to directly interact with cyclic nucleotides[10,50–52]. However, NMR and crystallographic structures later revealed that the residues making up this motif are not in direct contact with the cyclic nucleotide ligand. Nonetheless, the NKFDE motif was still deemed necessary for stabilizing the GAF domain fold, cyclic nucleotide binding, and transmission of the allosteric signal to the catalytic domain[2,13,17,50]. In fact, mutating each residue of the NKFDE motif individually into alanine in the PDE2 GAFb domain abolished cGMP binding[23]. A similar finding was observed upon mutating the D residue into alanine in the GAFa domain of PDE11A4[53]. Moreover, individually mutating N, K, and D residues of the NKFDE motif in PDE5 GAFa domain greatly decreased the cGMP binding affinity[50,54]. With that said, the role of the conserved NKFDE motif in cyclic nucleotide binding GAF domains remained enigmatic. Notably, F295 is the only residue amongst the cluster of the nine highly coevolving residue positions identified from our coevolutionary analysis (SCA) that is a part of the NKFDE motif. This critically highlights the significance of our study in understanding the ligand binding and allosteric regulation in GAF domain proteins. Overall, the results presented here provide insights into the mechanism underlying the allosteric regulation of GAF domains and could potentially pave the way for designing novel chemical modulators targeting GAF domain allostery.

## Conclusions

To conclude, we successfully utilized SCA to identify coevolving residue positions in GAF domains, two of which are distant from the ligand binding site yet play a critical role in ligand-binding induced conformational allostery. Our analysis suggests that these two residues affect the GAF domain-mediated allostery through different mechanisms. The L267 position appears to be primarily involved in controlling ligand traffic to (apo) and from (holo) the ligand binding site in the domain. On the other hand, the F295 position is important for maintaining the local inter-residue interactions, and its disruption likely affects both ligand binding (holo) and signal transduction to the catalytic domain that follows ligand binding to the GAF domain. As such, our study proposes: first, a ligand-binding model in GAF domains, in which ligand access to/from the binding site is controlled by the β3-α4 "lid" that gates the ligand binding site. Consequently, any factors that affect the stability or dynamics of these secondary structures are likely to affect the cGMP binding affinity. Second, we propose a ligand-binding induced signal transduction model, in which the conformational change signal that follows ligand binding is transmitted to the catalytic domain through the C-terminal helix (α5) in the domain. Importantly, our data suggest that factors disrupting signal transduction in the domain could have

serious detrimental effects on allostery that are comparable, if not more, to factors directly reducing ligand binding.

## Methods

### Identification of coevolving residues in GAF domains

To perform SCA[18,43], GAF domain protein sequence alignment was obtained from the Pfam database[20] and phylogenetically related sequences were removed using JalView[55] by applying a redundancy threshold value of 95%. The dataset was further refined by removing all positions with more than 20% gaps using available MATLAB scripts. SCA was performed using the algorithm described previously[19]. Hierarchical clustering analysis of the statistical coupling scores was performed by constructing a dendrogram of the correlation matrix using MATLAB. Cluster of residue positions with the highest statistical coupling scores were mapped on the PDE5 GAFa domain structure (PDB: 2K31[22]).

### MD simulation and trajectory analysis

MD simulations were performed using the available structure of apo (PDB: 3MF0)[16] or modeled structure of the holo GAFa domain of human PDE5. Homology modeling of holo GAFa domain of PDE5 was performed using the holo GAFa domain of mouse PDE5 (PDB: 2K31)[22] (Supplementary Fig. 10), which is the only available mammalian holo GAFa domain structure of PDE5 in the RCSB database, as a template. The model for the human holo GAFa domain of PDE5 (D164-I312) was generated using MODELLER (10.1 release, Mar. 18, 2021)[56], a widely used software package for homology modeling[57–64]. Briefly, the template (D154-E302) and human (D164-I312) GAFa domain sequences were aligned in the PIR format. This was followed by generating 10 structural models using MODELLER "Automodel" function and "very-slow" MD refining mode[56]. Assessment of the generated models was achieved using scoring functions such as DOPE, modpdf, and GA34. PROCHECK[65,66] was used to assess the stereochemical quality of the generated models (Supplementary Table 3). Quality evaluation of the chosen model was further assessed using the Ramachandran Plot (Supplementary Fig. 11) and the discrete optimized potential energy (DOPE) score profile (Supplementary Fig. 12). The alignment of the modeled GAFa domain with its template revealed that both share similar secondary structure elements.

For MD simulations, the structures of the mutants (Supplementary Data 1 to 6) were generated using the PyMOL mutagenesis tool. Parameter and topology files for the MD simulations were generated using the CHARMM-GUI server[67] and simulations were performed using the NAMD v2.13 software[68] and CHARMM36 force field[69] as described previously[70–74]. Structures were solvated using a TIP3P[71] cubic water box with a 10 Å minimum distance between the edge of the box and any of the biomolecular system atoms, and NaCl was added at a final concentration of 0.15 M with periodic boundary conditions applied. Before performing MD simulation production runs, energy minimization was first performed for 1000 timesteps, followed by a thermalization step where the systems were slowly heated to 310 K for 0.25 ns using a temperature ramp that raises the temperature at 1 K increment. The temperature was then maintained at 310 K using Langevin temperature control and at pressure of 1.0 atm using Nose-Hoover Langevin piston control. A constrained equilibration step of 1 ns was then performed where protein backbone atoms were constrained using a harmonic potential. Finally, 1000 ns MD simulation production runs were performed in triplicates for the WT and mutant systems. A 2 fs timestep of integration was chosen for all simulations where short-range non-bonded interactions were handled at 12 Å cut-off with 10 Å switching distance, while Particle-mesh Ewald (PME) scheme was used to handle long-range electrostatic interactions at 1 Å PME grid spacing. Analysis of the trajectories was performed using the available tools in the VMD software[75]. Dynamic cross-correlation (DCC) analysis was carried out as described previously[70,76] using the Bio3D package in R software[77]. Binding free energy changes were estimated utilizing the MM-PBSA method[78] through the CaFE 1.0 plugin[79] in the VMD software[75] using a Tcl script (Supplementary Text 2).

### Plasmid design and mutagenesis

The design of the BRET[2]-based GAFa and full-length PDE5 (A2 isoform) biosensors was described previously[24,25]. Briefly, in both biosensors, the GAFa domain (I159-N316) or the full-length PDE5A2, were flanked by the green fluorescent protein 2 (GFP[2]) at the N-terminal as the fluorescent energy acceptor, and by the *Renilla* luciferase (RLuc) at the C-terminal as the bioluminescent energy donor. In vitro, site-directed mutagenesis of the GAFa-based biosensor was performed using two mutagenic primers (5′ G ACA CAA AGC ATT GCC TGT ATG CCA ATT AAG 3′) and (5′ CA GGA AAC GGT GGT ACC GCT ACT GAA AAA GAT G 3′) to generate the L267A and F295A mutations in the GAFa domain, respectively, in a pBKS-GAFa plasmid. Mutations were confirmed through sequencing. The L267A and F295A mutant GAFa domain (residues I159-N316) sequences were then inserted into pGFP[2]-MCS-RLuc plasmid through restriction digestion using *Bgl*II and *Hind*III restriction enzymes and subsequent ligation to generate the GAFa-based mutant biosensor constructs (Supplementary Text 3)[24]. Mutations in the full-length PDE5A2 biosensor (pGPF[2]-PDE5A2-RLuc)[25] were generated by GenScript (GenScript, Singapore) (Supplementary Text 4).

To generate the mNeonGreen-GAFa-NanoLuc (mNG-GAFa-NLuc) biosensor construct, the GAFa sequence was synthesized and inserted into the pmNG-Rigidx3-NLuc plasmid using *Kpn*I-*Not*I restriction sites to generate the pmNG-GAFa-NLuc plasmid. For generating the miRFP670-nano3-picALuc(E50A) construct, the miRFPnano3 gene fragment was synthesized and inserted into the pmSca-picALuc(E50A)[34] using the *Eco*RI-*Bam*HI restriction sites to generate the pmiRFP670nano3-picALuc(E50A) plasmid. L267A and F295A mutants of pmNG-GAFa-NLuc, and L100A and W125A mutants of pmiRFP670nano3-picALuc(E50A) were then generated in their respective plasmids (GenScript, Singapore) (Supplementary Text 5, 6). A schematic describing the plasmids used in the current work is provided in Supplementary Fig. 13.

### Cell culture, transfection, and western blot analysis

The HEK 293 T cells were maintained in Dulbecco's Modified Eagle's Medium (DMEM) with 10% fetal calf serum (FCS), 120 mg/L penicillin and 270 mg/L streptomycin at 37 °C in a humidified incubator with an atmosphere of 5% $CO_2$. The cells were transfected with plasmids expressing the protein (WT or mutants) using either polyethyleneimine (PEI; Sigma-Aldrich, USA) or Lipofectamine 2000 (ThermoFisher Scientific, USA) transfection reagents according to the manufacturer's protocol.

For Western blot analysis, lysates prepared from HEK293T cells expressing the biosensor constructs were subjected to SDS-PAGE, and proteins were transferred to nitrocellulose membranes. Blots were probed overnight at 4 °C in TBS-T buffer containing 5% BSA with an anti-GFP antibody for GAFa and full-length PDE5 sensors (mouse IgG2a κ mAb, Santa Cruz Biotechnology, USA—SC-9996; 1:500 dilution), an anti-FLAG tag antibody for miRFP670nano3 constructs (mouse IgG2a mAB, Sino Biological, China—109143-MM13; 1:10000 dilution), or an anti-α-tubulin antibody for α-tubulin (mouse IgG2b mAB, Proteintech, USA—66031-1-Ig; 1:200000). The membranes were then washed with TBS-T, and probed with the same secondary antibody (anti-Mouse IgG:HRP Donkey pAb; ECM Biosciences, USA—MS3001; 1:20000 dilution) and developed using a chemiluminescent substrate in a ChemiDoc Imaging System (BIO-RAD, Germany).

### In vitro BRET assays

In vitro BRET assays were performed using cell lysates obtained from HEK293T cells transfected with plasmids expressing the GAFa domain-based (GFP[2]-GAFa(I159-N316)-RLuc) or the full-length PDE5 (GFP[2]-PDE5A2-RLuc) WT or mutant biosensor constructs[7,24–26,71]. Cells were washed in chilled Dulbecco's phosphate-buffered saline (DPBS) after 48 h of transfection, harvested using DPBS containing 2 mM EDTA, and lysed by sonication using an amplitude of 30% and a pulse of 10 s (3 cycles) in a lysis buffer containing 50 mM HEPES (pH 7.5), 100 mM NaCl, 2 mM ethylenediaminetetraacetic acid (EDTA), 1 mM dithiothreitol (DTT), 1× protease

inhibitor cocktail (ThermoFisher Scientific, USA) and 10% glycerol[24,26]. Sonicated lysates were centrifuged at 4 °C for 1 h at 18,400 g, and supernatants were collected.

All assays were performed at 37 °C in 96-well, white, flat-bottom plates. For bioluminescence and basal-BRET (i.e. without the addition of cGMP) measurements, 5 μL of the lysates were added to 40 μL of the lysis buffer, followed by the addition of 5 μL of 10x of RLuc substrate either coelenterazine 400a (for isolated GAFa-based assays) or Prolume purple (for full-length PDE5-based assays) (NanoLight Technology, Prolume Ltd., USA). For cGMP-binding induced BRET, 5 μL of the lysates were added to 35 μL of the lysis buffer and incubated with 5 μL of the indicated cGMP concentrations for 30 min, followed by the addition of a 5 μL of 10× of RLuc substrate either coelenterazine 400a (for isolated GAF domain biosensors) or Prolume purple (for the full-length PDE5A2 biosensors). Bioluminescence spectra of the WT or the mutant biosensors were acquired over wavelengths ranging between 385 and 665 nm utilizing a grating-based system with 15 nm integration step size, with each individual measurement acquired for 1 s using a Tecan SPARK® multimode microplate reader (Tecan, Switzerland). BRET was determined as a ratio of GFP[2] and RLuc emission, either in the absence or in the presence of the indicated concentrations of cGMP, following a 30-min incubation at 37 °C. Bioluminescence and BRET were measured following the addition of the RLuc substrates, either coelenterazine 400a or Prolume purple (NanoLight Technology, Prolume Ltd., USA).

### Determination of thermal stability of the GAFa domain of PDE5
Thermal stability of the WT and mutant GAFa domains was determined by measuring BRET from cell lysates obtained from HEK293T cells transfected with plasmids expressing the WT (mNG-GAFa-NLuc) or mutant (mNG-GAFa(L267A)-NLuc or mNG-GAFa(F295A)-NLuc) biosensors[30]. The assay was performed in 96-well, white, flat-bottom plates. Briefly, 1 μL of cell lysates were incubated in 45 μL of TBS buffer for 10 min at a range of temperatures (30.0, 31.4, 34.3, 38.9, 44.5, 49.3, 52.4, 60.0, 61.4, 63.9, 67.8, 72.3, 76.1, 78.8, and 80.0 °C) using a C1000 Touch Thermal Cycler (BIO-RAD, Germany), after which BRET was measured using GloMax® Discover plate reader (Promega, USA). BRET was measured following the addition of 5 μL of 10× of the substrate, furimazine (Promega, Wisconsin, USA), using a 450 nm band-pass filter for the donor (NLuc) and a 530 nm long-pass filter for the acceptor (mNG) and 0.4 s integration time. The BRET ratio was calculated by dividing the mNG emission by the NLuc emission. Data were fitted to a Boltzmann sigmoidal model to determine the melting temperature, $T_m$ (temperature at which the biosensor shows half maximal BRET). Three independent experiments were performed in triplicates.

### Determination of miRFP670nano3 fluorescence
Plasmids expressing the WT [pmiRFP670nano3-picALuc(E50A)] and mutant [pmiRFP670nano3(L100A)-picALuc(E50A) or pmiRFP670nano3(W125A)-picALuc(E50A)] constructs were transfected into HEK 293 T cells cultured into 96-well, white, flat-bottom plat, using polyethyleneimine (PEI; Sigma-Aldrich, Missouri, USA) transfecting reagent according to the manufacturers protocol and incubated for 48 h at 37 °C in a humidified incubator with an atmosphere of 5% $CO_2$. After 48 h of transfection, miRF670Pnano3 fluorescence was measured using GloMax® Discover plate reader (Promega, USA), utilizing a 627 nm excitation filter and a 660–720 emission filter, and picALuc bioluminescence intensity was determined following the addition of 50 μL of the substrate, coelenterazine h, utilizing Tecan SPARK® multimode microplate reader (Tecan, Switzerland). Five independent experiments were performed in triplicates.

### Data analysis and figure preparation
GraphPad Prism (version 8) and Microsoft Excel (2016) were used for data analysis and graphs preparation. The relative bioluminescence data were fitted using either a single (RLuc alone) or two Gaussian (GAFa domain biosensors and full-length biosensors) distributions. A dose-response curve

fitting was used for the log[cGMP] vs percentage change in BRET plot. Thermal stability data (change in BRET with changing temperature) were fitted using Boltzmann sigmoidal model. Figures were assembled using either Inkscape or Adobe Illustrator software.

### Reporting summary
Further information on research design is available in the Reporting Summary linked to this article.

## Data availability
The relevant data of this study are available within the manuscript and its supplementary files.

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

## Acknowledgements
This work is supported by a grant from HBKU Thematic Research Grant Program (VPR-TG01-007) and internal funding from the College of Health & Life Sciences, Hamad Bin Khalifa University, a member of the Qatar Foundation. W.S.A. is supported by a scholarship from CHLS, HBKU. A.T. was supported by a Summer Research Fellowship Program from the Indian Academy of Sciences. A.M.G and T.A. are supported by a postdoctoral fellowship from CHLS, HBKU. Some of the computational research work reported in the manuscript was performed using high-performance computer resources and services provided by the Research Computing group in Texas A&M University at Qatar. Research Computing is funded by the Qatar Foundation for Education, Science and Community Development (http://www.qf.org.qa).

## Author contributions
K.H.B. and S.S.V. conceived experiments. W.S.A., A.M.G., A.S., A.T., T.A., N.A., and K.H.B. performed experiments, analyzed data and prepared figure panels. W.S.A. and K.H.B. wrote the manuscript. All authors contributed to editing the manuscript and approved the manuscript.

## Funding

## Competing interests
The authors declare no competing interests.
