## [Peer Review File- corrected · Communications Chemistry]

Coevolving Residues Distant from the Ligand Binding Site are Involved in GAF Domain Function

Corresponding Author: Professor Kabir Biswas

Version 0:

Reviewer comments:

Reviewer #1

(Remarks to the Author)

The manuscript by Ahmed et al. seeks to understand the allosteric communication network in phosphodiesterase-5 (PDE5) that convey changes in conformation of the GAFa domain upon cGMP binding that ultimately result in acceleration of cGMP hydrolysis in the active site within the catalytic domain. The authors use statistical coupling analysis and MD simulations to identify potential allosteric “relay” sites distant from the cGMP binding pocket. They then generate site-directed mutants and rely on bioluminescence resonance energy transfer (BRET) measurements to measure ligand-induced conformational changes in both the isolated GAFa domain and full-length PDE5 for both wildtype and mutant recombinant proteins. In addition, they evaluate the likelihood that the allosteric communication network identified for cyclic nucleotide PDEs also pertains to a distantly related GAF domain that binds a completely different ligand.

The work represents a significant advance in our understanding of the allosteric regulation of PDE5, but also extends its significance by providing evidence that their widely distributed regulatory modules may rely on a conserved network of residues that convey signals from the ligand binding site to the protein domain to which the GAF domains are attached (e.g., the catalytic domain of the GAF-containing cyclic nucleotide phosphodiesterases). In most respects the results are well documented and support their conclusions. However, the impact of the manuscript is weakened by its length, the sequence of presentation of the results, and information that would enhance the clarity, rigor and reproducibility of the manuscript.

1. Sequence of presentation of results.

- a. Consider the following sequence that places the MD simulations after the biochemical results: (a) Identification of co-evolving residues using SCA; (a) BRET analysis of cGMP-induced conformational changes in WT and L267A and F295A GAFa domains; (c) BRET analysis of full-length PDE5 (WT and mutants); (d) evaluation of WT and mutant GAFa domain from cyanobacteria in its apo and liganded state; (d) MD simulations of the WT and mutant PDE5 GAFa domains; (e) identification of structural elements involved in allosteric communication upon cGMP binding.
- b. Consider omission of the thermal stability studies from the main text, as these data do not significantly add to the conclusions that can be drawn.

2. Materials and Methods section.

- a. Many of the plasmids were originally used in previous published works, and it is appropriate to only briefly describe them in this paper. However, the reader not familiar with the earlier papers would benefit from a supplementary figure in which the basic elements of each plasmid was visually depicted.
- b. L. 380: Please add more details here about the statistical tests and regression models used for curve fitting. The statistical tests are often included in the figure legends, but not the regression models used.
- c. Methods describing the biochemical assays lack information about the amount of protein (or number of cells) assayed in each method.

3. Results section.

- a. SCA is a powerful tool to identify coevolving residues that are likely of functional significance. The authors chose to use the GAF domain sequence alignment from the Pfam database, which includes GAF domains from all kingdoms. However, a subset of the entire family of GAF domains that contain two tandem GAF domains could have been chosen for SCA that might have identified a greater number of residues that coevolved along with the three high-scoring residues that were found from the entire GAF MSA. Please justify the choice to use an MSA of all known GAF-containing proteins and not just the subset of tandem GAF-containing proteins, since all of the GAF-containing PDEs have tandem GAF domains.
 - i. L 453-480: Justification for the exclusion of S289 from analysis should be provided by reporting in the legend of Fig. 2 the numerical coupling scores for the 9 co-evolving residues, since Fig. 2A has too low a resolution to identify individual residues and their scores.
 - b. Main Fig. 3 is very busy with too many small-sized panels. Some of these data could be moved to a supplemental figure,

with only a few selected panels being presented in the main manuscript.

c. Calculations for the MD simulations of the WT and mutant GAFa domains (Fig. 3A) are interpreted by the authors as revealing changes in structural dynamics, but the S.D. values for the RMSD values suggest that none of these changes are statistically significant. Furthermore, RMSD changes alone cannot suggest significant changes as the values are not uncommon for proteins of this size with flexible elements such as loops. In fact, if loops are excluded and the RMSD is recomputed, one may get even smaller values. Overall, one cannot easily claim whether apo or liganded GAF domain is more rigid.

i. Lines 509-525 should be revised to refer to many of the pairwise comparisons as “trends.” In contrast, the RMSF, Rg, residue distance, and SASA (Fig. 3B-F) are more significant and informative about the structural changes undergone by the three GAFa constructs. Note that the text (l. 635-640) inaccurately reports on the values shown in Fig. 3F.

ii. Interpretation of Fig. 3D (l. 551-571): the most obvious differences appear to be that WT undergoes a substantial decrease in Rg upon cGMP binding; the L267A mutant appears to be “locked” into the liganded conformation whereas the L295A mutant appears locked in the apo conformation. For both mutants, cGMP binding fails to induce a large change in Rg, unlike the WT.

iii. The authors should examine the sensitivity of their results to variation in force-field by conducting some additional simulations with the AMBER family of force-fields (at least one simulation of GAF in apo and liganded states if more is not possible). NAMD software allows conducting simulations with the AMBER FF. In case authors choose to not conduct these simulations, it should be added as a limitation somewhere in the discussion section.

d. The BRET results comparing the apo and liganded states of the GAFa domains of the WT and mutant constructs (Fig. 4) conclusively demonstrate that each of the mutants exhibits a large decrease in the binding affinity for cGMP compared with the WT GAFa domain which undergoes a significant increase in resonance energy transfer upon cGMP binding. In addition, the BRET assay enables estimation of the binding affinity for cGMP which is in close agreement with previous results from other labs for the WT GAFa construct. The two mutant GAFa constructs have ~10,000-fold reduced affinity, supporting the authors’ conclusion that these two residues, located at some distance from the cGMP binding pocket, are able to dramatically alter its conformation.

e. The BRET results for full-length PDE5 are in general consistent with the isolated GAFa domain, but the resonance energy transfer signals are considerably weaker—understandably so given that the two sensors are at a greater distance from each other in the holoenzyme. That said, the estimate for changes in cGMP binding affinity are more tenuous, given the 2-fold smaller amplitude for the WT holoenzyme relative to its GAFa domains (Fig. 4G vs. Fig. 5E), and the even smaller amplitude for the two full-length mutant enzymes (Fig. 5E). It is recommended to omit Fig. 5F for this reason, and report in the text only the good agreement of the EC50 values of the full-length WT and GAFa proteins. Likewise, Fig. 5G appears to be of tenuous significance given the large uncertainty in the EC50 values of the mutant full-length holoenzymes.

f. Figures 5H-J show results for a bacterial GAF domain that binds the ligand biliverdin. The primary significance of these results is to establish the universality of the allosteric communication between the GAFa ligand binding site and the coevolving residues homologous to L267 and F295. Figures 5H-K should be made into a separate figure.

g. Fig. 6 analyzes the probability distributions of two central structural elements that are proposed to function as a lid to the cGMP binding site ($\alpha 4$ - $\beta 3$) and an allosteric communication element ($\alpha 2$ - $\alpha 6$). These results support the prevailing model that the $\alpha 4$ - $\beta 3$ serves as a lid that closes upon cGMP binding to the WT GAFa domain. In contrast, analysis of the L267A mutant is consistent with the lid remaining open (hence, lowered binding affinity) and the allosteric activation region undergoing activation upon cGMP binding. In contrast, the F295 mutant exhibits more complex behavior than the other mutant, with allosteric signaling impaired in conjunction with reduced cGMP binding affinity. While this level of analysis is appropriate for focusing on these two structural elements, principal component analysis would offer a more comprehensive and robust approach to this issue.

4. Minor points:

a) Why was PDE5A2 chosen for study when PDE5A1 is the canonical sequence?

b) Use of the term “holo” to refer to the cGMP-bound state of PDE5 GAFa is somewhat misleading. “Holo” is an abbreviation of “holoenzyme,” which would refer to full-length, dimeric PDE5. A more accurate descriptor for the state where cGMP occupies the GAFa domain binding site might be “cGMP-bound PDE5” or “ligand-bound PDE5.”

c) L. 162: reference 53 should be reference 15.

d) Fig. 2A: provide a.a. numbers for x- and y-axes.

e) Supporting Fig. 1: It would be very helpful to add the α -helices and β -strands aligned above the primary sequence so that the structural elements extensively cited in the text could be associated with the residue numbering. See reference 16, Fig. 1 for an example.

f) The use of “ $\alpha 6$ ” to refer to the extension of the $\alpha 5$ helix is to be avoided. The standard GAF domain consists of only five α -helices (per ref. 16), one of which precedes the $\alpha 2$ helix resolved in the x-ray structures. The GAFa $\alpha 5$ helix is contiguous with the GAFa $\alpha 1$ helix.

g) Supporting Figure 2: red text within the plots is sometimes unreadable. Black font below figures needs to be larger, as it is barely readable.

h) Supporting Figure 3: the figure is too small, the x-axes are unreadable and there is no y-axis title. The zoomed in portion should be made easily readable because these are the highest scoring co-varying residues.

i) Fig. 4A and 4C are too small and the resolution is too low to be able to inspect the structural changes reported in the text. Larger images with greater pixel density are recommended. This comment also applies to some other structure images in the manuscript.

j) L 733: cite the papers reporting the 3MF0 and 2K31 structures that are being described in Fig. 4A at the end of this sentence.

k) L. 1899: why is mouse cGMP-bound 2K31 being used for the alignment with human apo 3MF0 when you have generated a structural model of human cGMP-bound GAFa?

l) L. 1903: In Fig. 4C, is the structure shown the 2K31 structure or the homology model?

- m) L. 2464 Supporting Fig. 9: the PDE2 GAF domain structure should be included, either 3IBJ or 1MC0.
- n) It is suggested that the authors provide pdb files for the structures that are shown in the movie files, and that the movie files be eliminated or reduced in number. Having the structure files would enable the reader to use visualization programs to view, manipulate, and compare the structural models reported in the manuscript.

Reviewer #2

(Remarks to the Author)

This work by Ahmed et al focuses on understanding how the function of a GAF domain is allosterically regulated by a pair of coevolved residues. A comprehensive MD analysis is performed and demonstrates convincingly that the mutations impact dynamics and cGMP binding in the GAFa domain. BRET experiments reveal that the mutations drastically affect binding of the cGMP to both individual domain and the full protein. The authors perform a detailed conformational analysis of their trajectories to propose mechanistic rationales for how these distant mutants achieve allosteric effects. This work is a valuable contribution because in addition to illuminating mechanisms of the GAF domain, it also provides a rigorous application of MD data to elucidate a molecular mechanism. Importantly, the authors show that the proposed allosteric effects here would apply more broadly to the GAF family. Because this is a coevolving residue pair, several questions come to mind, such as: what would be the functional impact of a double-mutant or a switched mutant (L267F/F295L). Admittedly these are outside the scope of this work but could make for interesting future investigations.

A few questions/unclear aspects are listed below, as well as some suggestions that could improve readership of the work.

what is the SCA score of the 2 coevolving residues? (I could not locate this information, apologies if I missed it). Also, can authors indicate on the heat map (Fig. 2A) the location of the residue pair that is the focus of this work? It could be nice if they can even zoom in on it to show these two positions.

This would probably not change the trend observed, but for the data in Fig 3E: distances between COM of coevolving positions, authors should consider using the C α distances instead of COM, which are different because L and F have been mutated to A at respective positions. The COM is quite heavily dependent on the size of the side chain. CA-CA distances might be a more equivalent comparison among the 3 complexes.

It is not clear to me how SASA of the entire GAF domain allows the authors to conclude that the ligand is less buried in the mutants compared to WT. To make this claim, authors would need to look specifically at SASA of amino acids involved in ligand binding and show that their SASA is increased. Higher SASA may just broadly indicate that parts of the protein have undergone unfolding, or a large conformational change.

The BRET results are very interesting. However, after reading the section (with the GAFa domain alone), I'm left with a very specific question: how exactly does the reduction in luminescence imply that a conformational change occurred in the mutants?

Additional explanation should be provided for the BRET experiment to help readers understand this. The authors provide a reference to a different paper where they have previously described this sensor, but it would be good for a reader to minimally be able to understand the basis of the experiment from reading this work without having to pull up a reference.

a bit confused by figure 5G. Text says that the x-axis represents the free energy measured from MD simulations. Were simulations also performed on the full-length proteins? If not, what do the blue triangles represent?

It was quite clunky reading through the description/locations of the 9 coevolving residues. Authors should consider simplifying by placing specific descriptions (e.g. located on sheet beta5, helix alpha3 etc) into a table, and keep the text uncluttered, e.g.: Positional mapping...revealed that six residues (V230, I266, C268, V281...) are located at or near the ligand binding site.

Numbers are mixed up in the SASA part of the manuscript (lines 637-640)

Reviewer #3

(Remarks to the Author)

In this manuscript, Ahmed et al. investigate the role of two coevolving residues within GAF domains, which are regulatory domains found in a wide range of proteins. Using sequence coevolution analysis (SCA), the authors identified coevolving residues in the GAF domain. They subsequently conducted a BRET experiment to evaluate the impact of mutations at these residues, specifically L267A and F295A. The results indicated that these mutations significantly reduced the binding affinity to cGMP compared to the wild type protein.

To explore the structural and dynamic effects of these mutations, the authors performed molecular dynamics (MD) simulations, which included dynamical cross-correlation (DCC) analyses between residues in both the wild type and mutant proteins.

Although the authors present an interesting observation that these two coevolving residues strongly influence ligand-binding

affinity, I find that the mechanistic explanation provided by the authors is insufficient and lacks clarity. Additionally, this manuscript requires major revisions to improve readability. In its current form, it is not suitable for publication in Communications Chemistry.

Below are some of the major concerns:

The significant impact of the L267A and F295A mutations suggests that they substantially alter the structure of the GAF domain, possibly by perturbing its hydrophobic core. The authors should experimentally verify the structural consequences of these mutations.

Stronger evidence is needed to demonstrate that the model of the holo structure of the GAF domain is sufficiently reliable for the MD simulations presented.

Regarding DCC analysis: the authors should provide a more detailed explanation of how changes in DCC between residues lead to a reduction in cGMP binding affinity. While the authors note that DCC between the mutation sites decreases, they fail to explain how this observation correlates with the observed decrease in binding affinity.

Lines 630-643: The authors state that the solvent-accessible surface area (SASA) of cGMP in the L267A and F295A mutants increased compared to the wild type. However, the values provided do not support this claim, and the SASA should be reported in units of angstrom^2 .

The authors argue that the mutations result in a more compact apo state based on the observation that the root mean square deviation (RMSD) and radius of gyration (Rg) are reduced compared to the wild type. However, the changes in these parameters are minimal, and even if significant, they do not adequately explain the reduction in binding affinity.

Line 1180: The authors suggest that the L267A mutation opens the $\alpha3$ - $\beta4$ lid in the holo form, thereby reducing binding affinity. However, they also claim that this is reflected by a decrease in ligand SASA in the L267A mutant compared to the wild type. This reasoning appears contradictory, as opening the binding site should increase the SASA, not decrease it.

Lines 1258-1261: I find no compelling evidence to support the argument presented in this section.

Version 1:

Reviewer comments:

Reviewer #1

(Remarks to the Author)
See attached file.

Reviewer #2

(Remarks to the Author)
All of my concerns have been addressed.

Reviewers' comments:

Reviewer #1 (Remarks to the Author):

Comment: The manuscript by Ahmed et al. seeks to understand the allosteric communication network in phosphodiesterase-5 (PDE5) that convey changes in conformation of the GAFa domain upon cGMP binding that ultimately result in acceleration of cGMP hydrolysis in the active site within the catalytic domain. The authors use statistical coupling analysis and MD simulations to identify potential allosteric “relay” sites distant from the cGMP binding pocket. They then generate site-directed mutants and rely on bioluminescence resonance energy transfer (BRET) measurements to measure ligand-induced conformational changes in both the isolated GAFa domain and full-length PDE5 for both wildtype and mutant recombinant proteins. In addition, they evaluate the likelihood that the allosteric communication network identified for cyclic nucleotide PDEs also pertains to a distantly related GAF domain that binds a completely different ligand.

The work represents a significant advance in our understanding of the allosteric regulation of PDE5, but also extends its significance by providing evidence that their widely distributed regulatory modules may rely on a conserved network of residues that convey signals from the ligand binding site to the protein domain to which the GAF domains are attached (e.g., the catalytic domain of the GAF-containing cyclic nucleotide phosphodiesterases). In most respects the results are well documented and support their conclusions. However, the impact of the manuscript is weakened by its length, the sequence of presentation of the results, and information that would enhance the clarity, rigor and reproducibility of the manuscript.

Response: We would like to thank the Reviewer for the excellent comments and suggestions on our manuscript. We have now addressed these in detail in our response below.

1. Sequence of presentation of results.

Comment: a. Consider the following sequence that places the MD simulations after the biochemical results: (a) Identification of co-evolving residues using SCA; (a) BRET analysis of cGMP-induced conformational changes in WT and L267A and F295A GAFa domains; (c) BRET analysis of full-length PDE5 (WT and mutants); (d) evaluation of WT and mutant GAFa domain from cyanobacteria in its apo and liganded state; (d) MD simulations of the WT and mutant PDE5 GAFa domains; (e) identification of structural elements involved in allosteric communication upon cGMP binding.

Response: We would like to thank the Reviewer for the suggestion. The current arrangement of the results serves the flow throughout the manuscript. Starting with the structural simulations lays the foundation for the hypothesis that mutating the coevolving residue positions affects the structure and ligand-binding induced conformational allostery in the domain, which we have explored in depth experimentally in the subsequent sections of the results section. We highly value the Reviewer’s suggestion while we opt to keep the current arrangement of the results section.

Comment: b. Consider omission of the thermal stability studies from the main text, as these data do not significantly add to the conclusions that can be drawn.

Response: We greatly appreciate the Reviewer’s suggestion. While the thermal stability data may not be at the core of the study, we still believe it provides valuable information on how the stability

of the domain is affected by mutating the coevolving residue positions. We greatly value the Reviewer's recommendation while we opt to keep these data in the manuscript as it addresses a key concern raised by other Reviewers on the impact of the mutations on the thermal stability of the GAF domain.

Comment: 2. Materials and Methods section.

a. Many of the plasmids were originally used in previous published works, and it is appropriate to only briefly describe them in this paper. However, the reader not familiar with the earlier papers would benefit from a supplementary figure in which the basic elements of each plasmid was visually depicted.

Response: We would like to thank the Reviewer for the suggestion. We have now included a new supplementary figure (Supplementary Figure 4) summarizing the plasmids used in the current work.

“

“Supporting Figure 4. Cartoon schematic depicting the plasmids used for generating the GAFa (left), full-length PDE5A2 (second left), miRFP670nano3 (second right), and GAFa thermal stability (right) WT and mutant protein constructs.”

The following statement has now been added to the “Plasmid design and mutagenesis” subsection of the methods section:

“A schematic describing the plasmids used in the current work is provided in Supporting Figure 4”

Comment: b. L. 380: Please add more details here about the statistical tests and regression models used for curve fitting. The statistical tests are often included in the figure legends, but not the regression models used.

Response: We appreciate the Reviewer's suggestion. We have now added more details about the statistical tests and regression models, including curve fitting, as the Reviewer has suggested.

“The relative bioluminescence data were fit using either a single (RLuc alone) or two Gaussian (GAFa domain biosensors and full-length biosensors) distributions. A dose-response curve fitting was used for the log[cGMP] vs percentage change in BRET plot. Thermal stability data (change in BRET with changing temperature) were fit using Boltzmann sigmoidal model.”

Comment: c. Methods describing the biochemical assays lack information about the amount of protein (or number of cells) assayed in each method.

Response: We would like to thank the Reviewer for the comment. We have now added more details describing the biochemical assays in the Materials and Methods section of the manuscript as described below:

“Cell culture, transfection, and western blot analysis

The HEK 293T cells were maintained in Dulbecco's Modified Eagle's Medium (DMEM) with 10% fetal calf serum (FCS), 120 mg/L penicillin and 270 mg/L streptomycin at 37 °C in a humidified incubator with an atmosphere of 5% CO₂. Before transfection, the HEK 293T cells were seeded onto 10 cm tissue culture dishes and incubated for 24 h pre-transfection. Post-incubation, the cells were checked for confluency (at 60-70%) and transfected with plasmids expressing the protein (WT or mutants) using either polyethylenimine (PEI; Sigma-Aldrich, USA) or Lipofectamine 2000 (ThermoFisher Scientific, USA) transfection reagents according to the manufacturer's protocols and incubated for 48 hrs. Untransfected cells and cells transfected with RLuc-only expressing plasmid were used as controls.

For Western blot analysis, equal volumes (30 µL) of cell lysates prepared from HEK 293T cells expressing the protein constructs (WT or mutants) were heated to 95 °C following the addition of Laemmli sample buffer (50 mM Tris-Cl pH 6.8, 1.6% SDS, 4% β-mercaptoethanol, 8% glycerol and 0.04% bromophenol blue). A 10% SDS-PAGE was used to separate the samples using running buffer (0.1% SDS, 192 mM glycine, 25 mM Tris) for constant voltage of 100 V for 1.5 h. The proteins were then transferred onto polyvinylidene fluoride membrane. The membranes were blocked at room temperature using Tris-buffered saline containing 0.1% Tween-20 (TBS-T) with bovine serum albumin (BSA; 5%) for 1 h. Blots were probed overnight at 4 °C in TBS-T buffer containing 5% BSA with an anti-GFP antibody for GAFa and full-length PDE5 sensors (mouse IgG2a κ mAb, Santa Cruz Biotechnology, USA – SC-9996; 1:500 dilution), an anti-FLAG tag antibody for miRFP670nano3 constructs (mouse IgG2a mAb, Sino Biological, China – 109143-MM13; 1:10000 dilution), or an anti-α-tubulin antibody for α-tubulin (mouse IgG2b mAb, Proteintech, USA – 66031-1; 1:200000). The membranes were then washed with TBS-T, and probed with the same secondary antibody (anti-Mouse IgG:HRP Donkey pAb; ECM Biosciences, USA – MS3001; 1:20000 dilution) and developed using a chemiluminescent substrate in a ChemiDoc Imaging System (BIO-RAD, Germany).

In vitro BRET assays

In vitro BRET assays were performed using cell lysates obtained from HEK293T cells transfected with plasmids expressing the GAFa domain-based (GFP²-GAFa(1159-N316)-RLuc) or the full-length PDE5 (GFP²-PDE5A2-RLuc) WT or mutant biosensor constructs [1-5]. Cells were washed in chilled Dulbecco's phosphate-buffered saline (DPBS) after 48 h of transfection, harvested using DPBS containing 2 mM EDTA, and lysed by sonication using an amplitude of 30% and a pulse of 10 sec (3 cycles) in a lysis buffer containing 50 mM HEPES (pH 7.5), 100 mM NaCl, 2 mM ethylenediaminetetraacetic acid (EDTA), 1 mM dithiothreitol (DTT), 1× protease inhibitor cocktail (ThermoFisher Scientific, USA) and 10% glycerol [1, 6]. Sonicated lysates were centrifuged at 4 °C for 1 h at 18,400g, and supernatants were collected.

All assays were performed at 37 °C in 96-well, white, flat-bottom plates. For bioluminescence and basal-BRET (i.e. without the addition of cGMP) measurements, 5 µL of the lysates were added to 40 µL of the lysis buffer, followed by the addition of 5 µL of 10x of RLuc substrate either coelenterazine 400a (for isolated GAFa-based assays) or Prolume purple (for full-length PDE5-based assays) (NanoLight Technology, Prolume Ltd., USA). For cGMP-binding induced BRET, 5 µL of the lysates were added to 40 µL of the lysis buffer and indicated with varying concentrations of cGMP for 30 min, followed by the addition of a 5 µL of 10x of RLuc substrate either coelenterazine 400a (for isolated GAF domain biosensors) or Prolume purple (for the full-length PDE5A2 biosensors). Bioluminescence spectra of the WT or the mutant biosensors were acquired over wavelengths ranging between 385 to 665 nm utilizing a grating-based system with 15 nm

integration step size, with each individual measurement acquired for 1 second using a Tecan SPARK® multimode microplate reader (Tecan, Switzerland). BRET was determined as a ratio of GFP² and RLuc emission, either in the absence or in the presence of the indicated concentrations of cGMP, following a 30-minute incubation at 37 °C. Bioluminescence and BRET were measured following the addition of the RLuc substrates, either coelenterazine 400a or Prolume purple (NanoLight Technology, Prolume Ltd., USA).

Determination of thermal stability of the GAFa domain of PDE5

Thermal stability of the WT and mutant GAFa domains was determined by measuring BRET from cell lysates obtained from HEK293T cells transfected with plasmids expressing the WT (mNG-GAFa-NLuc) or mutant (mNG-GAFa(L267A)-NLuc or mNG-GAFa(F295A)-NLuc) biosensors.[7] The assay was performed in a 96-well, white, flat-bottom plates. Briefly, 1 µL of cell lysates were incubated in 45 µL of TBS buffer for 10 min at a range of temperatures (30.0, 31.4, 34.3, 38.9, 44.5, 49.3, 52.4, 60.0, 61.4, 63.9, 67.8, 72.3, 76.1, 78.8, 80.0 °C) using a C1000 Touch Thermal Cycler (BIO-RAD, Germany), after which BRET was measured using GloMax® Discover plate reader (Promega, USA). BRET was measured following the addition of 5 µL of the substrate, furimazine (Promega, Wisconsin, USA), using a 450 nm band-pass filter for the donor (NLuc) and a 530 nm long-pass filter for the acceptor (mNG) and 0.4 s integration time. The BRET ratio was calculated by dividing the mNG emission over the NLuc emission. Data were fit to a Boltzmann sigmoidal model to determine the melting temperature, T_m (temperature at which the biosensor shows half maximal BRET). Three independent experiments were performed in triplicates.

Determination of miRFP670nano3 fluorescence

Plasmids expressing the WT [pmiRFP670nano3-picALuc(E50A)] and mutant [pmiRFP670nano3(L100A)-picALuc(E50A) or pmiRFP670nano3(W125A)-picALuc(E50A)] constructs were transfected into HEK 293T cells cultured into 96-well, white, flat-bottom plat, using polyethyleneimine (PEI; Sigma-Aldrich, Missouri, USA) transfecting reagent according to the manufacturers protocol and incubated for 48 h at 37 °C in a humidified incubator with an atmosphere of 5% CO₂. After 48 h of transfection, miRFP670Pnano3 fluorescence was measured using GloMax® Discover plate reader (Promega, USA), utilizing a 627 nm excitation filter and a 660-720 emission filter, and picALuc bioluminescence intensity was determined following the addition of 50 µL of the substrate, coelenterazine h, utilizing Tecan SPARK® multimode microplate reader (Tecan, Switzerland). Five independent experiments were performed in triplicates.”

3. Results section:

Comment: a. SCA is a powerful tool to identify coevolving residues that are likely of functional significance. The authors chose to use the GAF domain sequence alignment from the Pfam database, which includes GAF domains from all kingdoms. However, a subset of the entire family of GAF domains that contain two tandem GAF domains could have been chosen for SCA that might have identified a greater number of residues that coevolved along with the three high-scoring residues that were found from the entire GAF MSA. Please justify the choice to use an MSA of all known GAF-containing proteins and not just the subset of tandem GAF-containing proteins, since all of the GAF-containing PDEs have tandem GAF domains.

Response: We would like to thank the Reviewer for the comment. We wanted to have a global look at the coevolution of amino acid positions in the GAF domain family. We agree that GAF domains in humans and other organisms are present as tandem GAF domains in selected PDEs. However, this may not be the case in other species, such as the cyanobacteriochrome photoreceptors, *Anabaena* Adenylyl cyclase, and cyanobacterium *Nostoc* sp. PCC7120 proteins (such as all 4978). The GAF domain in these examples is not found in tandem. We have now clarified this in the Introduction section:

“While some GAF domains are found in tandem (e.g. the GAF domains of mammalian PDEs) [8], others are not (e.g. the GAF domain of cyanobacteriochrome photoreceptors and Anabaena Adenylyl cyclase)”

Comment: i. L 453-480: Justification for the exclusion of S289 from analysis should be provided by reporting in the legend of Fig. 2 the numerical coupling scores for the 9 co-evolving residues, since Fig. 2A has too low a resolution to identify individual residues and their scores.

Response: We would like to thank the Reviewer for the suggestion. In addition to the L276 and F295 coevolving residue positions, the S289 residue is also positioned away from the ligand binding site. However, in the GAF α domain of PDE5, S289 is located on a flexible loop (β 6- α 5 loop) and is not involved in any inter-residue interactions in both the WT and mutant GAF α domain of PDE5 (please refer to the residue interaction map below created using Arpeggio webserver [9]). We were more interested in the other two coevolving residues (i.e. L267 and F295) because they formed several inter-residue interactions, including between each other (Supporting Table 5 and Supporting Figure 13).

Per the Reviewer’s suggestion, we have now pinpointed the location of the nine coevolving residue positions (Fig. 2A) and highlighted the area in the heatmap where they intersect (white box). We have updated Fig. 2 and its legend accordingly.

Fig. 2. SCA reveals a cluster of coevolving residues in GAF domains. (A) Color-coded heat map showing pairwise scores of individual residue positions in GAF domains obtained from SCA. The positions of the nine coevolving residues are indicated. The area in the heatmap where the coevolving residue positions intersect is highlighted by a white box. (B) Cartoon representation of the GAFa domain of PDE5 (PDB: 2K31[10]) showing the coevolving cluster of residues (spheres in light blue) in the GAF domains. (C) Cartoon representation of the GAFa domain of PDE5 showing residues (side chains as sticks in pink) within a 5 Å distance from cGMP (left panel), coevolving residues (spheres in light pink; rest of the coevolving residues in light blue) within 5 Å distance from cGMP (middle panel) and coevolving residues L267 and F295 selected for further functional characterization (dark blue spheres, right panel).

Comment: b. Main Fig. 3 is very busy with too many small-sized panels. Some of these data could be moved to a supplemental figure, with only a few selected panels being presented in the main manuscript.

Response: We appreciate the Reviewer's concern and agree with it in that it is busy. However, we would like to mention that we have made an effort to capture all key results from the MD simulation analysis of the WT and two mutant GAF domains, in the apo and the holo forms. As such, we are comparing six different MD simulation trajectories and therefore, there won't be much increase in the size of the figure panels even if we move some of the panels to the Supporting Information section. Again, we highly appreciate the Reviewer's feedback.

Comment: c. Calculations for the MD simulations of the WT and mutant GAFa domains (Fig. 3A)

are interpreted by the authors as revealing changes in structural dynamics, but the S.D. values for the RMSD values suggest that none of these changes are statistically significant. Furthermore, RMSD changes alone cannot suggest significant changes as the values are not uncommon for proteins of this size with flexible elements such as loops. In fact, if loops are excluded and the RMSD is recomputed, one may get even smaller values. Overall, one cannot easily claim whether apo or liganded GAF domain is more rigid.

Response: We greatly appreciate the Reviewer's insight on this and completely agree with the Reviewer on this. The RMSD values provide a global view of the changes in the structural dynamics of the GAFa domain over the simulation time and indicate if the simulations have reached some sort of 'equilibrium'. The indicated S.D. values were calculated for the entire simulation time (including the initial phase starting from zero angstrom). The loops are important elements of the domain that we believe would be difficult to exclude from the analysis. We agree with the Reviewer that loops, by their nature, would be expected to have higher structural dynamics compared to other structural elements, and we did not claim GAF domain rigidity or flexibility based solely on the RMSD values. Instead, we combined this with other computational analyses such as RMSF, Rg, DCC, and ligand SASA to get more perspective on the structural changes of the domain. Hence, we agree with the Reviewer that the RMSD values alone are not enough to conclude the structural changes or claim rigidity/flexibility of the domain.

Comment: i. Lines 509-525 should be revised to refer to many of the pairwise comparisons as "trends." In contrast, the RMSF, Rg, residue distance, and SASA (Fig. 3B-F) are more significant and informative about the structural changes undergone by the three GAFa constructs. Note that the text (l. 635-640) inaccurately reports on the values shown in Fig. 3F.

Response: We thank the Reviewer for this comment and agree with it. We have now referred to the pairwise comparisons of the RMSD values in the indicated lines as "trends".

"We observed the following trends in the RMSD measurements."

On the other hand, the RMSF, Rg, residue distances, and SASA are all discussed in detail in lines 615-637, 639-664, 666-696, and 723-736, respectively.

We thank the Reviewer for highlighting the reporting inaccuracy. We have now corrected the ligand SASA values reporting in these lines of the main text as follows:

"This analysis revealed large increases in the cGMP SASA in both the L267A ($135.4 \pm 143.5 \text{ \AA}^2$) and F295A ($163.6 \pm 150.8 \text{ \AA}^2$) mutant GAFa domains, as compared to the WT GAFa domain ($38.8 \pm 28.2 \text{ \AA}^2$) (Fig. 3F)."

Comment: ii. Interpretation of Fig. 3D (l. 551-571): the most obvious differences appear to be that WT undergoes a substantial decrease in Rg upon cGMP binding; the L267A mutant appears to be "locked" into the liganded conformation whereas the L295A mutant appears locked in the apo conformation. For both mutants, cGMP binding fails to induce a large change in Rg, unlike the WT.

Response: We would like to thank the Reviewer for this valuable insight. We agree with the Reviewer, and we have now indicated that in the results section:

"This decrease appears similar to that observed in holo WT domain, potentially suggesting the apo L267A mutant to be 'locked' in the liganded form. However, in the holo form, the F295A mutant showed an increase

in the R_g compared to holo WT, while this increase was not reflected in the holo L267A mutant (14.6 ± 0.2 , 14.9 ± 0.2 and 15.3 ± 0.4 Å for the holo WT, L267A and F295A mutant domains, respectively) (Fig. 3D), potentially suggesting the holo F295A mutant to be 'locked' in the apo form.”

Comment: iii. The authors should examine the sensitivity of their results to variation in force-field by conducting some additional simulations with the AMBER family of force-fields (at least one simulation of GAF in apo and liganded states if more is not possible). NAMD software allows conducting simulations with the AMBER FF. In case authors choose to not conduct these simulations, it should be added as a limitation someplace in the discussion section.

Response: We would like to thank the Reviewer for this suggestion and agree with it. However, we would like to mention that the current work did not intend to examine the sensitivity of the biomolecular simulation system to different force fields. However, this could be explored in future research. We have now clarified this in the discussion section:

“We note that the current work did not intend to examine the effect of different force fields on the biomolecular simulation systems, such as AMBER FF, which could be explored in future work.”

Comment: d. The BRET results comparing the apo and liganded states of the GAFa domains of the WT and mutant constructs (Fig. 4) conclusively demonstrate that each of the mutants exhibits a large decrease in the binding affinity for cGMP compared with the WT GAFa domain which undergoes a significant increase in resonance energy transfer upon cGMP binding. In addition, the BRET assay enables estimation of the binding affinity for cGMP which is in close agreement with previous results from other labs for the WT GAFa construct. The two mutant GAFa constructs have ~10,000-fold reduced affinity, supporting the authors' conclusion that these two residues, located at some distance from the cGMP binding pocket, are able to dramatically alter its conformation.

Response: We would like to thank the Reviewer for thoroughly summarizing the results in Fig. 4.

Comment: e. The BRET results for full-length PDE5 are in general consistent with the isolated GAFa domain, but the resonance energy transfer signals are considerably weaker—understandably so given that the two sensors are at a greater distance from each other in the holoenzyme. That said, the estimate for changes in cGMP binding affinity are more tenuous, given the 2-fold smaller amplitude for the WT holoenzyme relative to its GAFa domains (Fig. 4G vs. Fig. 5E), and the even smaller amplitude for the two full-length mutant enzymes (Fig. 5E). It is recommended to omit Fig. 5F for this reason, and report in the text only the good agreement of the EC50 values of the full-length WT and GAFa proteins. Likewise, Fig. 5G appears to be of tenuous significance given the large uncertainty in the EC50 values of the mutant full-length holoenzymes.

Response: We would like to thank the Reviewer for the insight and suggestion. We agree that the amplitude of change in BRET in the case of the full-length protein is less than that those reported for the isolated GAFa domain. However, this is expected as the full-length biosensor reports the change in conformation in the entire protein while the isolated GAFa domain biosensor reports the conformational change in the isolated GAFa domain only. Therefore, there is no contradiction between the results of the isolated GAFa domain and the full-length PDE5.

Comment: f. Figures 5H-J show results for a bacterial GAF domain that binds the ligand biliverdin. The primary significance of these results is to establish the universality of the allosteric communication between the GAFa ligand binding site and the coevolving residues homologous to L267 and F295. Figures 5H-K should be made into a separate figure.

Response: We thank the reviewer for the suggestion. We have now split Figure 5H-K into a separate figure (Fig. 6) and edited the figure citations in the text.

Comment: g. Fig. 6 analyzes the probability distributions of two central structural elements that are proposed to function as a lid to the cGMP binding site ($\alpha 4$ - $\beta 3$) and an allosteric communication element ($\alpha 2$ - $\alpha 6$). These results support the prevailing model that the $\alpha 4$ - $\beta 3$ serves as a lid that closes upon cGMP binding to the WT GAF α domain. In contrast, analysis of the L267A mutant is consistent with the lid remaining open (hence, lowered binding affinity) and the allosteric activation region undergoing activation upon cGMP binding. In contrast, the F295 mutant exhibits more complex behavior than the other mutant, with allosteric signaling impaired in conjunction with reduced cGMP binding affinity. While this level of analysis is appropriate for focusing on these two structural elements, principal component analysis would offer a more comprehensive and robust approach to this issue.

Response: We would like to thank the Reviewer for this suggestion. We have now performed PCA (average of three trajectories) and find that the GAF domains show multiple clusters in the PC1 vs. PC2 plots in the presence of the cGMP ligand, holo form, unlike the apo form. However, we did not find any appreciable difference between the WT and mutant complexes and, therefore, decided not to include the results in the manuscript. Please find the results of the PCA below:

Legend: Plots of PC1 vs. PC2 obtained from PCA of the three MD simulation trajectories (average) of the WT, L267A and F295A mutant GAF domains in the apo and holo forms.

4. Minor points:

Comment: a) Why was PDE5A2 chosen for study when PDE5A1 is the canonical sequence?

Response: We would like to thank the Reviewer for the question. We want to clarify that all PDE5A isoforms (PDE5A1, PDE5A2, and PDE5A3) share the same amino acid sequence, including the tandem GAF domains and the catalytic domain, and only differ in the length of the N-terminal region. Therefore, choosing any of the PDE5 isoforms is unlikely to have an impact on the current investigation and the reported results. We used the PDE5A2 isoform for feasibility, as we have already developed an in-house biosensor that characterizes the structural changes in the full-length protein [11].

Comment: b) Use of the term “holo” to refer to the cGMP-bound state of PDE5 GAF α is somewhat misleading. “Holo” is an abbreviation of “holoenzyme,” which would refer to full-length, dimeric PDE5. A more accurate descriptor for the state where cGMP occupies the GAF α domain binding site might be “cGMP-bound PDE5” or “ligand-bound PDE5.”

Response: We thank the Reviewer for the comment. We would like to point out that the terms “apo” and “holo” states are clarified (Page 5) to refer to the unliganded and cGMP-bound GAF α domain of PDE5, respectively, throughout the manuscript:

“the three-dimensional structures of the isolated PDE5 GAFa domain both in the absence of cGMP (unliganded; apo) and cGMP-bound (holo) forms are available”

Comment: c) L. 162: reference 53 should be reference 15.

Response: We would like to thank the Reviewer and mention that we have now corrected this in the manuscript.

Comment: d) Fig. 2A: provide a.a. numbers for x- and y-axes.

Response: We would like to thank the Reviewer for this suggestion. We have now edited Fig. 2A and provided the amino acid numbers for the x- and y-axis. We have also indicated the positions of the nine coevolving amino acid residues and edited the figure legend.

“**Fig. 2. SCA reveals a cluster of coevolving residues in GAF domains.** (A) Color-coded heat map showing pairwise scores of individual residue positions in GAF domains obtained from SCA. The positions of the nine coevolving residues are indicated. The area in the heatmap where the coevolving residue positions intersect is highlighted by a white box. (B) Cartoon representation of the GAFa domain of PDE5 (PDB: 2K31[10]) showing the coevolving cluster of residues (spheres in light blue) in the GAF domains. (C) Cartoon representation of the GAFa domain of PDE5 showing residues (side chains as sticks in pink) within a 5 Å distance from cGMP (left panel), coevolving residues (spheres in light blue) within 5 Å distance from cGMP (middle panel) and coevolving residues L267 and F295 selected for further functional characterization (dark blue spheres, right panel).”

Comment: e) Supporting Fig. 1: It would be very helpful to add the α -helices and β -strands aligned above the primary sequence so that the structural elements extensively cited in the text could be associated with the residue numbering. See reference 16, Fig. 1 for an example.

Response: We would like to thank the Reviewer for this suggestion. As suggested, we have included the secondary structure elements in Supporting Figure 1 (please see below).

Supporting Figure 1. Amino acid sequence alignment of mouse and human PDE5 GAFa domain. Differences in the amino acid sequences are highlighted in red. **Secondary structure elements (α helices and β sheets)** are indicated above the sequences.

Comment: f) The use of “ $\alpha 6$ ” to refer to the extension of the $\alpha 5$ helix is to be avoided. The standard GAF domain consists of only five α -helices (per ref. 16), one of which precedes the $\alpha 2$ helix resolved in the x-ray structures. The GAFa $\alpha 5$ helix is contiguous with the GAFa $\alpha 1$ helix.

Response: We would like to thank the Reviewer for this comment. We agree with the Reviewer, and we have now corrected the typo in the main text as well as in Fig. 6.

Comment: g) Supporting Figure 2: red text within the plots is sometimes unreadable. Black font below figures needs to be larger, as it is barely readable.

Response: We would like to thank the Reviewer for this suggestion. As suggested, we have now increased the font size in the plot statistics and also indicated the name of the residues in the red font (generously allows and disallowed regions) in the plot statistics.

Comment: h) Supporting Figure 3: the figure is too small, the x-axes are unreadable and there is no y-axis title. The zoomed in portion should be made easily readable because these are the highest scoring co-varying residues.

Response: We would like to thank the Reviewer for this suggestion. We have now added axis titles and clarified the coevolving residue positions on the zoom-in panel.

Comment: i) Fig. 4A and 4C are too small and the resolution is too low to be able to inspect the structural changes reported in the text. Larger images with greater pixel density are recommended. This comment also applies to some other structure images in the manuscript.

Response: We would like to thank the Reviewer for this comment. We have now included higher-resolution images in Fig. 4A and 4C as suggested.

Comment: j) L 733: cite the papers reporting the 3MF0 and 2K31 structures that are being described in Fig. 4A at the end of this sentence.

Response: We would like to thank the Reviewer for this suggestion. We have now cited the two references in the main text as suggested by the Reviewer.

Comment: k) L. 1899: why is mouse cGMP-bound 2K31 being used for the alignment with human apo 3MF0 when you have generated a structural model of human cGMP-bound GAFa?

Response: We would like to thank the Reviewer for this question. The alignment shows the amino acid differences between the human PDE5 GAFa and the mouse model of the cGMP-bound GAFa of PDE5 (PDB ID: 2K31), which we used as a template for the homology modeling of the human cGMP-bound (holo-form) GAFa domain of PDE5. As we noted in the manuscript, the cGMP-bound GAFa domain of mouse PDE5 is the only available structural model of a cGMP-bound PDE5 GAFa domain in the RCSB database. We have now clarified this in detail in the manuscript:

“Homology modeling of holo GAFa domain of PDE5 was performed using the holo GAFa domain of mouse PDE5 (PDB: 2K31) [10] (Supporting Figure 1), which is the only available mammalian holo GAFa domain structure of PDE5 in the RCSB database, as a template. The model for the human holo GAFa domain of PDE5 (D164-I312) was generated using MODELLER (10.1 release, Mar. 18, 2021) [12], a widely used software package for homology modeling [13-20]. Briefly, the template (D154-E302) and human (D164-I312) GAFa domain sequences were aligned in the PIR format. This was followed by generating 10 structural models using MODELLER “Automodel” function and “very-slow” MD refining mode.[12] Assessment of the generated models was achieved using scoring functions such as DOPE, modpdf, and GA34. PROCHECK [21, 22] was used to assess the stereochemical quality of the generated models (Supporting Table 1). Quality evaluation of the chosen model was further assessed using the Ramachandran Plot (Supporting Figure 2) and discrete optimized potential energy (DOPE) score profile (Supporting Figure 3).”

Comment: l) L. 1903: In Fig. 4C, is the structure shown the 2K31 structure or the homology model?

Response: We would like to thank the Reviewer for this question. The structure shown is for the mouse PDE5 GAFa (2K31), which shares similar structural elements to those of the modeled human PDE5 GAFa domain. For the Reviewer’s convenience, we are sharing below the structural alignment between the 2K31 (gray) and the modeled human holo-GAFa domain (white).

Comment: m) L. 2464 Supporting Fig. 9: the PDE2 GAF domain structure should be included, either 3IBJ or 1MC0.

Response: We would like to thank the Reviewer for this suggestion. We have now added the structure of GAFb domain of PDE2 (3IBJ, below) in the Supporting Figure suggested by the Reviewer:

Comment: n) It is suggested that the authors provide pdb files for the structures that are shown in the movie files, and that the movie files be eliminated or reduced in number. Having the structure files would enable the reader to use visualization programs to view, manipulate, and compare the structural models reported in the manuscript.

Response: We would like to thank the Reviewer for this comment. We have now provided the pdb files for the apo and holo, WT and mutant, GAFa domains in the Supporting Information section.

Reviewer #2 (Remarks to the Author):

Comment: This work by Ahmed et al focuses on understanding how the function of a GAF domain is allosterically regulated by a pair of coevolved residues. A comprehensive MD analysis is performed and demonstrates convincingly that the mutations impact dynamics and cGMP binding

in the GAFa domain. BRET experiments reveal that the mutations drastically affect binding of the cGMP to both individual domain and the full protein. The authors perform a detailed conformational analysis of their trajectories to propose mechanistic rationales for how these distant mutants achieve allosteric effects. This work is a valuable contribution because in addition to illuminating mechanisms of the GAF domain, it also provides a rigorous application of MD data to elucidate a molecular mechanism. Importantly, the authors show that the proposed allosteric effects here would apply more broadly to the GAF family. Because this is a coevolving residue pair, several questions come to mind, such as: what would be the functional impact of a double-mutant or a switched mutant (L267F/F295L). Admittedly these are outside the scope of this work but could make for interesting future investigations.

Response: We thank the Reviewer for succinctly summarizing our findings. Based on our current results, we expect the impact of mutating the two residue positions to have a more drastic effect on the structure and function of the domain. We agree that exploring the impact of the switch mutants would be interesting. However, we note that the two coevolving residue positions not only form hydrophobic interaction with each other but also with other neighboring residues (Supporting Table 5 and Supporting Figure 13). Therefore, we expect the switch mutant to behave differently from the wild-type domain. Nonetheless, this could be the subject of future investigations.

Comment: A few questions/unclear aspects are listed below, as well as some suggestions that could improve readership of the work.

Response: We would like to thank the Reviewer for the valuable comments and suggestions. We have now replied to these in detail.

Comment: what is the SCA score of the 2 coevolving residues? (I could not locate this information, apologies if I missed it). Also, can authors indicate on the heat map (Fig. 2A) the location of the residue pair that is the focus of this work? It could be nice if they can even zoom in on it to show these two positions.

Response: We would like to thank the Reviewer for the suggestion. We have now indicated the position of the nine coevolving residue positions on the heatmap (Fig. 2A) and highlighted the intersection area of the nine coevolving residue positions (white box).

Comment: This would probably not change the trend observed, but for the data in Fig 3E: distances between COM of coevolving positions, authors should consider using the C-alpha distances instead of COM, which are different because L and F have been mutated to A at respective positions. The COM is quite heavily dependent on the size of the side chain. CA-CA distances might be a more equivalent comparison among the 3 complexes.

Response: We thank the Reviewer for this insightful comment. We would like to mention that we specifically used the COM distance, rather than the C-alpha distance, to see how the interaction between the two residues is altered across the simulation time. As the C-alpha distance calculation excludes the side chains, it may reflect less accurately the interaction distance between the two residues. Nonetheless, when performing distance calculations using C-alpha atoms (please see the figure below), we did not observe a substantial difference from the trend observed by measuring COM distances in Fig. 3E.

Comment: It is not clear to me how SASA of the entire GAF domain allows the authors to conclude that the ligand is less buried in the mutants compared to WT. To make this claim, authors would

need to look specifically at SASA of amino acids involved in ligand binding and show that their SASA is increased. Higher SASA may just broadly indicate that parts of the protein have undergone unfolding, or a large conformational change.

Response: We thank the author for the comment. We would like to clarify here that the calculated SASA in Fig 3F is for the ligand and not the protein, as indicated in the text and in Fig. 3F., with higher values suggesting ‘loose’ positioning of the ligand, cGMP, in the binding site.

Comment: The BRET results are very interesting. However, after reading the section (with the GAFa domain alone), I’m left with a very specific question: how exactly does the reduction in luminescence imply that a conformational change occurred in the mutants?

Additional explanation should be provided for the BRET experiment to help readers understand this. The authors provide a reference to a different paper where they have previously described this sensor, but it would be good for a reader to minimally be able to understand the basis of the experiment from reading this work without having to pull up a reference.

Response: We would like to thank the Reviewer for the comment. The BRET signal is dependent on multiple factors including the spectral overlap between the two reporters, the distance between them, and the spatial orientation of the two reporters with respect to each other. Since the two reporters are the same for the WT and mutant sensors, any change in the BRET signal would suggest a change in the distance and/or orientation of the protein reporters, which can only happen if there is a conformational change in the domain. Therefore, the change in the basal BRET signal (in the absence of added cGMP ligand) observed in the mutants suggests a conformational change in the protein as a result of the mutations. While a change in the cGMP dose-response (i.e. BRET) curve would suggest a change in the cGMP-binding induced conformational change in the domain/protein. We have now clarified this in the manuscript on page 5 and page 8:

“Third, we have previously engineered BRET-based conformational biosensors for both the isolated GAFa domain and the full-length PDE5. These biosensors faithfully report conformational changes in the proteins that occur as a result of mutation (basal BRET) or as a consequence of ligand-binding (cGMP-binding induced BRET), which has enabled the characterization of several mutations with regards to GAF domain allostery.”

“BRET is a biophysical technique that involves resonance energy transfer between a bioluminescent donor and a fluorescent acceptor, the extent of which is dependent on the spectral overlap, distance, and relative orientation between the two reporters. In this case, the biosensor consists of an isolated GAFa domain of PDE5 sandwiched between GFP² at the N-terminal side (BRET acceptor) and RLuc at the C-terminal side (BRET donor). GFP² is a brighter variant of GFP with a F64L mutation and has an emission spectra similar to that of GFP but with significantly blue-shifted excitation spectra with a peak of 396 nm (Patent no. US 6.12,188 B1) [23], leading to a substantial spectral overlap of RLuc emission with GFP² excitation as well as a large separation of RLuc and GFP² emissions [1, 2, 6]. Since the two protein reporters are the same for the WT and mutant sensors, then any change in the BRET signal would suggest a change in the distance and/or orientation of the protein reporters, which can only happen if there is a conformational change in the domain as a result of the mutation (change in basal BRET) or ligand-binding (change in cGMP-binding induced BRET).”

Comment: a bit confused by figure 5G. Text says that the x-axis represents the free energy measured from MD simulations. Were simulations also performed on the full-length proteins? If not, what do the blue triangles represent?

Response: We would like to thank the Reviewer for raising this concern. We note that we performed

BRET-based biosensor assays using GAFa domain-based biosensors (WT and mutants) and full-length PDE5 biosensors (WT and mutants). In the indicated plot, we plotted the $\log(EC_{50})$ of the cGMP-binding induced conformational change (y-axis), in both the isolated and full-length PDE5, against the free energy of binding of cGMP to GAFa domain (x-axis) obtained from MD simulation trajectories of the isolated holo-GAFa domain. In response to the concern raised by the Reviewer, we would like to clarify that the blue triangles represent the EC_{50} of cGMP-binding induced conformational change in the full-length PDE5 plotted against the free energy of binding of cGMP to GAFa domain (x-axis) obtained from MD simulation trajectories of the isolated holo-GAFa domain. We have now clarified this in the legend of Fig. 5G:

“Fig. 5. Mutation of distant, coevolving residues alters cGMP-induced allosteric regulation in the full-length PDE5. (A) Schematic showing the mechanism of the BRET²-based, full-length PDE5A2 conformational biosensor in the absence and presence of cGMP. (B) Western blot analysis of the cell lysates prepared from HEK293T cells transfected with either the WT or mutant full-length PDE5A2 biosensor plasmid constructs probed using an anti-GFP antibody showing the expression of biosensor constructs. (C) Graphs showing bioluminescence spectra obtained from lysates prepared from cells expressing the WT and mutant L267A and F295A mutant PDE5 biosensor constructs. Note the reduction in the GFP² emission peaks in the L267A and F295A mutant PDE5 biosensors. Data shown are mean \pm S.D. from a representative experiment, with experiments performed three times. (D) Bar graph showing the basal BRET values of the WT, L267A, and F295A mutant PDE5 biosensors. Note the significant reduction in the basal BRET values of the L267A and F295A mutants compared to the WT GAFa domain. (E) Graphs showing a percentage decrease in BRET values of the WT, L267A, and F295A mutant PDE5 biosensors after 30 min incubation with the indicated cGMP concentrations. Note the decrease in the maximum %change in BRET as well as the rightward shift in the cGMP dose-response curves of the mutant biosensors. (F) Graph showing $\log(EC_{50})$ values of cGMP-induced conformational change for the WT, L267A and F295A mutant GAFa domains in the full-length PDE5. Inset, values on top indicate the EC_{50} of cGMP-induced conformational change for the respective GAFa domains in the full-length PDE5 biosensor. For (D), (E), and (F), data shown are mean \pm S.D. from three experiments, with each experiment performed in triplicates. (G) Graph showing $\log(EC_{50})$ (mean \pm S.D.) values of cGMP-binding induced conformational change in the **isolated** GAFa domain and full-length PDE5 **BRET-based biosensors** against ΔG (mean \pm S.D.) values obtained from the MD simulation runs of **the isolated** GAFa domain (WT and mutants).”

Comment: It was quite clunky reading through the description/locations of the 9 coevolving residues. Authors should consider simplifying by placing specific descriptions (e.g. located on sheet beta5, helix alpha3 etc) into a table, and keep the text uncluttered, e.g.: Positional mapping...revealed that six residues (V230, I266, C268, V281...) are located at or near the ligand binding site.

Response: We would like to thank the Reviewer for the suggestion. We have now simplified the text as follows:

“Positional mapping of the nine coevolving residue positions (Fig. 2B, Supporting Table 1) revealed that six residues (V230, I266, C268, V281, Q283, I285; all residue numbers indicated are as per human PDE5A1 sequence) are located in (V230, I266, Q283, I285) or near (C268, V281) the ligand binding site in the GAFa domain (binding site residues were identified using a cutoff of 5 Å [24] distance from cGMP in the holo GAFa domain [10]). On the other hand, the remaining three residues (S289, L267, and F295) are located distant from the binding site with no direct interaction with the cGMP (Fig. 2C, Supporting Movie 1).”

We have also added a supporting table (Supporting Table 1) indicating the location of the nine coevolving residue positions:

“**Supporting Table 1:** Positional mapping of the coevolving residues on the GAFa domain of PDE5A1.”

Coevolving residues	Position
V230	α 3
I266	β 5
C268	β 5
V281	β 6
Q283	β 6
I285	β 6
S289	β 6- α 5 loop
L267	β 5
F295	β 6- α 5 loop

Comment: Numbers are mixed up in the SASA part of the manuscript (lines 637-640)

Response: We would like to thank the Reviewer for highlighting the inaccuracy in the indicated text. We have now corrected the SASA values reporting in these lines of the main text:

“This analysis revealed large increases in the cGMP SASA in both the L267A ($135.4 \pm 143.5 \text{ \AA}^2$) and F295A ($163.6 \pm 150.8 \text{ \AA}^2$) mutant GAFa domains, as compared to the WT GAFa domain ($38.8 \pm 28.2 \text{ \AA}^2$) (Fig. 3F).”

Reviewer #3 (Remarks to the Author):

Comment: In this manuscript, Ahmed et al. investigate the role of two coevolving residues within GAF domains, which are regulatory domains found in a wide range of proteins. Using sequence coevolution analysis (SCA), the authors identified coevolving residues in the GAF domain. They subsequently conducted a BRET experiment to evaluate the impact of mutations at these residues, specifically L267A and F295A. The results indicated that these mutations significantly reduced the binding affinity to cGMP compared to the wild type protein.

To explore the structural and dynamic effects of these mutations, the authors performed molecular dynamics (MD) simulations, which included dynamical cross-correlation (DCC) analyses between residues in both the wild type and mutant proteins.

Although the authors present an interesting observation that these two coevolving residues strongly influence ligand-binding affinity, I find that the mechanistic explanation provided by the authors is insufficient and lacks clarity. Additionally, this manuscript requires major revisions to improve readability. In its current form, it is not suitable for publication in Communications Chemistry. Below are some of the major concerns:

Response: We would like to thank the Reviewer for their comments and suggestions. We have now addressed these in detail in our response below.

Comment: The significant impact of the L267A and F295A mutations suggests that they substantially alter the structure of the GAF domain, possibly by perturbing its hydrophobic core.

The authors should experimentally verify the structural consequences of these mutations.

Response: We would like to thank the Reviewer for this comment and agree with the Reviewer in that the mutations can substantially alter the structure of the GAF domain. However, we would like to mention that our computational and experimental analyses clearly show that the two mutations lead to some structural changes in the domain, as could be ascertained from the altered (reduced) basal BRET ratio of the biosensor. Importantly, thermal stability measurements revealed small, although statistically significant, decreases in the melting temperature of the two mutants compared to the WT GAF domain. Further, we would like to mention that we found these structural changes to alter, but not completely abolish, the cGMP-binding induced conformational regulation of the GAF, as ascertained from the rightward shift in the cGMP concentration-dependent BRET ratio increases. Furthermore, a detailed structural analysis of the WT and the mutant GAF domains revealed that L267 and F295 mutations disrupt the local hydrophobic interaction between these residues and neighboring residues (Supporting Table 5), which, in turn, likely affects cGMP binding and subsequent allosteric signal transmission to the terminal $\alpha 5$ helix of the domain. An overview of the structural changes associated with two mutations has been provided in Fig. 3 and in lines 573-818 in the main text. More detailed insights into the structural changes induced by these mutations are provided in lines 1038-1238 and Fig. 6.

Comment: Stronger evidence is needed to demonstrate that the model of the holo structure of the GAF domain is sufficiently reliable for the MD simulations presented.

Response: We would like to thank the Reviewer for this comment. The 3D structure of the cGMP-bound GAF α domain of human PDE5 is not currently available. Therefore, we performed homology modeling of holo GAF α domain of human PDE5 using the cGMP-bound GAF domain of mouse PDE5, which is readily available in the PDB database (PDB ID: 2K31). We generated 10 homology models using MODELLER [25], a widely used software for homology modeling [13-20], and is generally highly reliable for generating structural models of highly homologous proteins. As can be seen from the sequence alignment of the two proteins (human and mouse) presented in Supporting Figure 1, there are only 5 residues, out of a total 148, that are different between the human and the mouse protein.

Supporting Figure 1. Amino acid sequence alignment of mouse and human PDE5 GAF α domain. Differences in the amino acid sequences are highlighted in red. Secondary structure elements (α helices and β sheets) are indicated above the sequences.

Further, assessment of the generated models was achieved using scoring functions such as DOPE, modpdf, and GA341. The stereochemical quality was assessed using PROCHECK (Supporting

Figure 2). We have added more details in the text regarding the method used for homology modeling and quality assessment of the generated models:

“Homology modeling of holo GAFa domain of PDE5 was performed using the holo GAFa domain of mouse PDE5 (PDB: 2K31) [10] (Supporting Figure 1), which is the only available mammalian holo GAFa domain structure of PDE5 in the RCSB database, as a template. The model for the human holo GAFa domain of PDE5 (D164-I312) was generated using MODELLER (10.1 release, Mar. 18, 2021) [12], a widely used software package for homology modeling [13-20]. Briefly, the template (D154-E302) and human (D164-I312) GAFa domain sequences were aligned in the PIR format. This was followed by generating 10 structural models using MODELLER “Automodel” function and “very-slow” MD refining mode.[12] Assessment of the generated models was achieved using scoring functions such as DOPE, molpdf, and GA34. PROCHECK [21, 22] was used to assess the stereochemical quality of the generated models (Supporting Table 1). Quality evaluation of the chosen model was further assessed using the Ramachandran Plot (Supporting Figure 2) and discrete optimized potential energy (DOPE) score profile (Supporting Figure 3).”

We have included an additional supporting table illustrating the model evaluation (Supporting Table 1). The model that was chosen for further MD simulation analysis (highlighted in green) was selected based on the score assessment since it has the lowest molpdf and DOPE scores, and the max GA341 score. It also has the most residues in the allowed region of the Ramachandran plot.

Supporting Table 1. List of the holo GAFa domain models of human PDE5 that were generated through homology modeling holo GAFa domain of mouse PDE5 as a template (PDB ID: 2K31). The model chosen for MD simulation investigation (highlighted in green) has the lowest molpdf and DOPE scores, and the max GA341 score among the generated models. It also has the highest number of residues in the core region of the Ramachandran plot.

Model	Score assessment			Ramachandran plot (%)				RMSD
	molpdf	DOPE	GA341	core	allow	generous	disallow	
Holo_GAFa_model.B99990001.pdb	828	-14487	1	92.5	6	1.5	0	0.387
Holo_GAFa_model.B99990002.pdb	715	-14717	1	94	3.8	2.3	0	0.377
Holo_GAFa_model.B99990003.pdb	791	-14524	1	93.2	4.5	1.5	0.8	0.37
Holo_GAFa_model.B99990004.pdb	824	-14478	1	93.2	4.5	1.5	0.8	0.395
Holo_GAFa_model.B99990005.pdb	789	-14399	1	93.2	4.5	1.5	0.8	0.374
Holo_GAFa_model.B99990006.pdb	885	-14360	1	93.2	4.5	2.3	0	0.403
Holo_GAFa_model.B99990007.pdb	825	-14671	1	93.2	4.5	2.3	0	0.391
Holo_GAFa_model.B99990008.pdb	760	-14431	1	92.5	6	1.5	0	0.406
Holo_GAFa_model.B99990009.pdb	760	-14599	1	92.5	4.5	2.3	0.8	0.366
Holo_GAFa_model.B99990010.pdb	752	-14510	1	93.2	4.5	2.3	0	0.381
2k31_NL.pdb (template)				88	9	2.3	0.8	0

The quality of the chosen model was assessed using the Ramachandran Plot displayed in Supporting Figure 2 (below):

Based on an analysis of 118 structures of resolution of at least 2.0 Angstroms and R-factor no greater than 20%, a good quality model would be expected to have over 90% in the most favoured regions.

Based on an analysis of 118 structures of resolution of at least 2.0 Angstroms and R-factor no greater than 20%, a good quality model would be expected to have over 90% in the most favoured regions.

Supporting Figure 2. Ramachandran plot of the modeled holo GAFa domain of human PDE5. The structural model of the human GAFa domain of PDE5 (left panel) spanning residues from D164 to I312 was generated from the mouse holo PDE5-GAFa model (PDB: 2K31) (right panel). The core regions (red) represent the most favorable combination of phi-psi values. The percentage of residues in the core is 94%, depicting a better stereochemical quality of the modeled structure compared to the mouse template structure, which has 88% of residues in the core region.

We have also included an additional supporting figure highlighting the DOPE profiling of the Model vs Template (Supporting Figure 3):

Supporting Figure 3. Quality Estimation of the modeled holo GAFa domain of human PDE5 (Red) compared to the modeling template GAF domain of mouse PDE5 (Green) using discrete optimized potential energy (DOPE) score profiling.

Comment: Regarding DCC analysis: the authors should provide a more detailed explanation of how changes in DCC between residues lead to a reduction in cGMP binding affinity. While the authors note that DCC between the mutation sites decreases, they fail to explain how this observation correlates with the observed decrease in binding affinity.

Response: We would like to thank the Reviewer for this suggestion. The rationale behind performing the DCC analysis is to provide additional supporting evidence that mutating the two positions in the GAF domain alters the dynamically cross-correlated motions both in the whole domain (Supporting Figure 7) as well as with respect to the two positions (Fig. 3G), which may contribute to the observed change in the domain's function, including ligand binding. Specific to the point raised by the Reviewer, the decrease in dynamically cross-correlated motions that we reported in the results between the coevolving residues and sheet β_4 in the mutants compared to the WT domain could be attributed to the fact that the β_4 carries the 267 residue position, and the disruption of the local inter-residue interaction between this position and surrounding residues as a result of mutating this position or the 295 position (Supporting Table 5 and Supporting Figure 13) contributes to the decrease in the DCC motions and may have an effect on ligand-binding induced conformational allostery. We have now clarified this in the text:

“A decrease in the correlated motions of residue positions 267 and 295 with residues in sheets β_4 , which holds the 267 position, was observed in the apo L267A and F295A mutant, as compared to the WT, GAFa domain. It is possible that disruption in the inter-residue interaction caused by mutating the 267 and 295 positions results in the observed decrease in DCC motions between these positions and sheet β_4 structure, which may have an effect on ligand binding.”

Comment: Lines 630-643: The authors state that the solvent-accessible surface area (SASA) of

cGMP in the L267A and F295A mutants increased compared to the wild type. However, the values provided do not support this claim, and the SASA should be reported in units of \AA^2 .

Response: We thank the Reviewer for pointing this out. The units were mistakenly indicated as \AA . We have now corrected this and reported the values in \AA^2 :

“This analysis revealed large increases in the cGMP SASA in both the L267A ($135.4 \pm 143.5 \text{\AA}^2$) and F295A ($163.6 \pm 150.8 \text{\AA}^2$) mutant GAFa domains, as compared to the WT GAFa domain $38.8 \pm 28.2 \text{\AA}^2$ (Fig. 3F).”

Comment: The authors argue that the mutations result in a more compact apo state based on the observation that the root mean square deviation (RMSD) and radius of gyration (Rg) are reduced compared to the wild type. However, the changes in these parameters are minimal, and even if significant, they do not adequately explain the reduction in binding affinity.

Response: We would like to thank the Reviewer for the comment. We performed RMSD and Rg analyses to see how mutating the two positions changes the overall dynamics (RMSD) and compactness (Rg) of the domain, the changes of which could ultimately play a role in the domain's function, including ligand binding. These analyses are not meant to directly explain ligand binding. However, the most obvious differences appear to be that WT undergoes a substantial decrease in Rg upon cGMP binding; the apo L267A mutant appears to be “locked” into the liganded conformation whereas the holo L295A mutant appears locked in the apo conformation. We have now indicated this in the results section:

“This decrease appears similar to that observed in holo WT domain, potentially suggesting the apo L267A mutant to be locked in the liganded form. However, in the holo form, the F295A mutant showed an increase in the Rg compared to holo WT, while this increase was not reflected in the holo L267A mutant (14.6 ± 0.2 , 14.9 ± 0.2 and $15.3 \pm 0.4 \text{\AA}$ for the holo WT, L267A and F295A mutant domains, respectively) (Fig. 3D), potentially suggesting the holo F295A mutant to be locked in the apo form.”

More importantly, the detailed explanation of how the structural changes in the domain affect ligand binding and allostery is provided in Fig. 6 and lines 1038-1238.

Comment: Line 1180: The authors suggest that the L267A mutation opens the $\alpha 3$ - $\beta 4$ lid in the holo form, thereby reducing binding affinity. However, they also claim that this is reflected by a decrease in ligand SASA in the L267A mutant compared to the wild type. This reasoning appears contradictory, as opening the binding site should increase the SASA, not decrease it.

Response: We thank the Reviewer for pointing this out. This was a typo on our part. We have now corrected this in the text:

“Additionally, our MD simulation analysis showed that, in the holo form, the L267A mutant results in the opening of the $\alpha 3$ - $\beta 4$ “lid” that gate the cGMP binding site, making it difficult to retain the ligand in the binding site. This was reflected by an increase in the ligand SASA, and a decrease in interaction energy, hydrogen bond formation, and binding free energy in the L267A mutant domain compared to the WT.”

Comment: Lines 1258-1261: I find no compelling evidence to support the argument presented in this section.

Response: We would like to thank the Reviewer for the comment. This conclusion was based on the increase of the calculated $\log(\text{EC}_{50})$ of the ligand-binding induced conformational change in

the full-length PDE5 as shown in Fig. 5F, which is statistically significant for the mutants compared to the WT.

References:

1. Biswas, K., S. Sopory, and S. Visweswariah, *The GAF domain of the cGMP-binding, cGMP-specific phosphodiesterase (PDE5) is a sensor and a sink for cGMP*. *Biochemistry*, 2008. **47**(11): p. 3534.
2. Biswas, K. and S. Visweswariah, *Distinct allostery induced in the cyclic GMP-binding, cyclic GMP-specific phosphodiesterase (PDE5) by cyclic GMP, sildenafil, and metal ions*. *The Journal of biological chemistry*, 2011. **286**(10): p. 8545.
3. Biswas, K.H. and S.S. Visweswariah, *Buffer NaCl concentration regulates Renilla luciferase activity and ligand-induced conformational changes in the BRET-based PDE5 sensor*. *Matters*, 2017. **10.19185/matters.201702000015**.
4. Biswas, K.H., et al., *Cyclic nucleotide binding and structural changes in the isolated GAF domain of Anabaena adenylyl cyclase, CyaB2*. *PeerJ*, 2015. **3**: p. e882.
5. Altamash, T., et al., *Intracellular Ionic Strength Sensing Using NanoLuc*. *International journal of molecular sciences*, 2021. **22**(2): p. E677.
6. Biswas, K.H. and S.S. Visweswariah, *Buffer NaCl concentration regulates Renilla luciferase activity and ligand-induced conformational changes in the BRET-based PDE5 sensor*. *Matters*, 2017. **3**(5): p. e201702000015.
7. Hoare, B.L., et al., *ThermoBRET: a ligand-engagement nanoscale thermostability assay applied to GPCRs*. *bioRxiv*, 2020: p. 2020.08. 05.237982.
8. Ahmed, W.S., A.M. Geethakumari, and K.H. Biswas, *Phosphodiesterase 5 (PDE5): Structure-function regulation and therapeutic applications of inhibitors*. *Biomedicine & pharmacotherapy= Biomedecine & pharmacotherapie*, 2021. **134**: p. 111128.
9. Jubb, H.C., et al., *Arpeggio: A Web Server for Calculating and Visualising Interatomic Interactions in Protein Structures*. *Journal of Molecular Biology*, 2017. **429**(3): p. 365.
10. Heikaus, C.C., et al., *Solution structure of the cGMP binding GAF domain from phosphodiesterase 5: insights into nucleotide specificity, dimerization, and cGMP-dependent conformational change*. *J Biol Chem*, 2008. **283**(33): p. 22749-59.
11. Biswas, K.H. and S.S. Visweswariah, *Distinct allostery induced in the cyclic GMP-binding, cyclic GMP-specific phosphodiesterase (PDE5) by cyclic GMP, sildenafil, and metal ions*. *Journal of Biological Chemistry*, 2011. **286**(10): p. 8545-54.
12. Eswar, N., et al., *Protein structure modeling with MODELLER*. *Methods Mol Biol*, 2008. **426**: p. 145-59.
13. Jeong, H., et al., *Structures of the TMC-1 complex illuminate mechanosensory transduction*. *Nature*, 2022. **610**(7933): p. 796-803.
14. Singh, S., et al., *Nucleolar maturation of the human small subunit processome*. *Science (New York, NY)*, 2021. **373**(6560): p. eabj5338.
15. Wang, B., et al., *The pesticide chlorpyrifos promotes obesity by inhibiting diet-induced thermogenesis in brown adipose tissue*. *Nature communications*, 2021. **12**(1): p. 5163.
16. Evans, S.W., et al., *A positively tuned voltage indicator for extended electrical recordings in the brain*. *Nature methods*, 2023. **20**(7): p. 1104-1113.
17. Thomasen, F.E., et al., *Rescaling protein-protein interactions improves Martini 3 for flexible proteins in solution*. *Nature communications*, 2024. **15**(1): p. 6645.

18. Dolton, G., et al., *Targeting of multiple tumor-associated antigens by individual T cell receptors during successful cancer immunotherapy*. Cell, 2023. **186**(16): p. 3333-3349. e27.
19. Oh, M., et al., *The lipoprotein-associated phospholipase A2 inhibitor Darapladib sensitises cancer cells to ferroptosis by remodelling lipid metabolism*. Nature communications, 2023. **14**(1): p. 5728.
20. Xu, Z., et al., *Ligand recognition and G-protein coupling of trace amine receptor TAAR1*. Nature, 2023. **624**(7992): p. 672-681.
21. Laskowski, R., M. MacArthur, and J. Thornton, *PROCHECK: validation of protein-structure coordinates*. 2012.
22. Laskowski, R., et al., *PROCHECK: a program to check the stereochemical quality of protein structures*. Journal of Applied Crystallography, 1993. **26**(2): p. 283-291.
23. Zimmermann, T., et al., *Spectral Imaging and Linear Un-Mixing Enables Improved FRET Efficiency With a Novel GFP2-YFP FRET Pair*. FEBS letters, 2002. **531**(2): p. 245-249.
24. Zhang, Q., et al., *Probing the Molecular Mechanism of Rifampin Resistance Caused by the Point Mutations S456L and D441V on Mycobacterium Tuberculosis RNA Polymerase Through Gaussian Accelerated Molecular Dynamics Simulation*. Antimicrobial agents and chemotherapy, 2020. **64**(7): p. e02476-19.
25. Webb, B. and A. Sali, *Protein Structure Modeling with MODELLER*. Methods in molecular biology (Clifton, NJ), 2021. **2199**: p. 239-255.

Reviewer #1 (Remarks to the Author):

Comment: In the revised manuscript by Ahmed et al., the authors have only partially addressed the concerns raised by the three reviewers. In general, the authors' refusal to address some of the global issues about the readability and organization of the manuscript (cited by Reviewers 1 and 3) was disappointing and borderline disrespectful of the time and energy the reviewers spent providing commentary. In addition, there remain specific issues raised by the reviewers that were adequately clarified by the authors, but not incorporated into the revised manuscript. The specific comments below reference the page number of the "Reviewers' comments" document, and the comment number/letter on each page.

Response: We sincerely thank the Reviewer for their time and effort in reviewing our manuscript and for providing thoughtful follow-up comments. We have addressed each of these points in detail in our response.

Comment: p.1 Comment #1a and #1b: The refusal to reorganize the manuscript as suggested is adequately justified by the authors' responses.

Response: We would like to thank the Reviewer for understanding and acknowledging our justification regarding the manuscript structure. We appreciate the opportunity to explain our reasoning and are glad it has been found satisfactory by the Reviewer.

Comment: p. 3-4 Comment c.: The request was for additional information on the biochemical assays, but the authors have added significant new details in the "Cell culture, transfection, and western blot analysis" section that were not requested and add further to the length of the manuscript. The information about the BRET assays, thermal stability studies, and fluorescence would have sufficed.

Response: We would like to thank the Reviewer for their suggestion regarding the additional details included in the "Cell Culture, Transfection, and Western Blot Analysis" section. We understand the concern about the manuscript's length and have now streamlined this section by removing extraneous details. We have now focused the edits only on the requested information about the BRET assays, thermal stability studies, and fluorescence. We have removed the previously added details from the "Cell culture, transfection, and western blot analysis" section.

Comment: p. 4 Comment a.: The authors make the case that their approach of examining a single GAF domain is of greater relevance for an understanding of the entire GAF domain family. While this is indeed true, the fact is that the GAF domains found in PDE5 exist in tandem, and that the allosteric communication pathway is initiated in the cyclic nucleotide binding domain in GAFa and but is communicated through the GAFb domain to the catalytic domain. The authors need to acknowledge in the manuscript the limitations of studying only PDE5 GAFa by stating that it does not address how GAFa structure and ligand-induced conformational changes are altered when GAFb is absent.

Response: We would like to thank the Reviewer for sharing this concern. It is indeed well-documented that the conformational change induced in the GAFa domain by cGMP binding is

transmitted to the catalytic domain through the GAFb domain in PDE5. Our aim was to investigate how the impact of mutating the two coevolving residue positions observed in the isolated GAFa domain translates to the full-length PDE5 as an endpoint. We agree with the Reviewer that the allosteric signal is transmitted through the GAFb domain and that removing the GAFb domain is expected to affect signal transmission to the catalytic domain. We have now addressed this in the revised manuscript in the Results section, pages 13 and 14, as follows:

“It is important to note that the ligand-induced conformational change initiated in the GAFa domain is transmitted to the catalytic domain of PDE5 through the GAFb domain. Consequently, this allosteric communication is expected to be altered in the absence of the GAFb domain. While this is true, our focus was to investigate how changes in ligand-induced conformational allostery observed in the isolated, mutant GAFa domains translate to the full-length PDE5 as an endpoint.”

Comment: p. 6 Comment b.: Authors have dismissed the suggestion to improve the readability of the manuscript by relegating some of the panels in Fig. 3 to Supplementary Information. Their reply is reasonable, but reflects a pattern of not recognizing the issue of “readability.”

Response: **Response:** We appreciate the Reviewer’s feedback on our response to the suggestion in the first round of review.

Comment: p. 6-7 Comment c.: Another example where the authors agree with the criticism, but don’t alter the manuscript by eliminating the sentences that are being critiqued.

Response: We would like to thank the Reviewer for the input. We have now added the following to the Results section of the manuscript on page 7:

“Notably, RMSD measurements alone may not provide sufficient information to conclude or compare the changes in structural dynamics induced by mutating the two coevolving residue positions. Therefore, to gain deeper insights into the structural changes in the GAFa domain in the apo and holo states, we performed additional trajectory analyses, including RMSF, radius of gyration (R_g), DCC, and ligand SASA.”

Comment: p. 8 Comment iii: Since the scientific premise for the biochemical experiments is based on the MD simulations, it is reasonable to request that at least one comparison of results between AMBER and CHARMM force fields be conducted. The authors’ addition to the manuscript is acceptable, but part of the larger pattern of deflecting reasonable critiques of the work.

Response: We appreciate the Reviewer’s feedback on our response to the suggestion in the first round of review. While a comparison of different force fields is desirable, we would like to submit that we are limited by available resources to perform additional 9000 ns of explicit solvent, all atom MD simulation.

Comment: p. 8 Comment e.: The authors’ response is in agreement with the reviewer’s observation that the amplitude of the full-length protein (10-20% change) is less than for the GAFa domain (~50% change). However, the authors fail to address the much greater uncertainty in calculating the EC50 for the smaller amplitude exhibited by the full-length protein. The recommendation remains

that Fig. 5E should be presented, but that Fig. 5F is of questionable value and should be omitted. [It seems odd that the WT vs L267A has a >10-fold lower p value than the WT vs F295A given the large difference in EC50 values of the two mutants relative to the WT.]

Response: We appreciate the Reviewer's insight into this. As suggested by the Reviewer, we have removed panel F from Fig. 5 and edited the figure legend accordingly.

“Fig. 5. Mutation of distant, coevolving residues alters cGMP-induced allosteric regulation in the full-length PDE5. (A) Schematic showing the mechanism of the BRET²-based, full-length PDE5A2 conformational biosensor in the absence and presence of cGMP. (B) Western blot analysis of the cell lysates prepared from HEK293T cells transfected with either the WT or mutant full-length PDE5A2 biosensor plasmid constructs probed using an anti-GFP antibody showing the expression of biosensor constructs. (C) Graphs showing bioluminescence spectra obtained from lysates prepared from cells expressing the WT and mutant L267A and F295A mutant PDE5 biosensor constructs. Note the reduction in the GFP² emission peaks in the L267A and F295A mutant PDE5 biosensors. Data shown are mean \pm S.D. from a representative experiment, with experiments performed three times. (D) Bar graph showing the

basal BRET values of the WT, L267A, and F295A mutant PDE5 biosensors. Note the significant reduction in the basal BRET values of the L267A and F295A mutants compared to the WT GAFa domain. (E) Graphs showing a percentage decrease in BRET values of the WT, L267A, and F295A mutant PDE5 biosensors after 30 min incubation with the indicated cGMP concentrations. Note the decrease in the maximum %change in BRET as well as the rightward shift in the cGMP dose-response curves of the mutant biosensors. (F) Graph showing log(EC₅₀) values of cGMP-induced conformational change for the WT, L267A and F295A mutant GAFa domains in the full-length PDE5. Inset, values on top indicating the EC₅₀ of cGMP-induced conformational change for the respective GAFa domains in the full-length PDE5 biosensor. For (D) and (E), and (F), data shown are mean ± S.D. from three experiments, with each experiment performed in triplicates. (G) Graph showing log(EC₅₀) (mean ± S.D.) values of cGMP-induced conformational change in the GAFa domain and full-length PDE5 proteins against ΔG (mean ± S.D.) values obtained from the MD simulation runs of GAFa domain (WT and mutants).”

Comment: p. 9 Comment 4. Minor points, a.: The authors misinterpreted the reviewer’s concern that the N-terminal region preceding the GAFa domain could influence the allosteric behavior of the PDE5 GAFa domain. PDE5A1 has additional amino acids at its N-terminus compared to PDE5A2. Wang et al (2010; ref. 16) has shown that the N-terminal sequence preceding GAFa folds over onto the GAFa domain, potentially interacting with and affecting allosteric communication. Giorgi et al. (2023) more recently reported that the N-terminal region affects the state of oligomerization of PDE5. While the reviewer understands the rationale the author use for working with the PDE5A2 sequence, the manuscript should address the possibility that the choice of isoform could have important implications for understanding the allosteric regulation of PDE5 holoenzyme.

Response: We would like to thank the Reviewer for their invaluable insight into this. Indeed, it has been suggested that the N-terminal loop plays a role in cGMP-binding induced allostery [1]. We have now indicated this in the revised manuscript in the Results section, page 11:

“We note that the N-terminal loop of PDE5 has been suggested to play a role in the cGMP-induced conformational allostery [1]. Therefore, the choice of PDE5 isoform could have important implications for understanding the allosteric regulation of PDE5 catalytic activity, as the PDE5A2 isoform, which we utilized in the design of the full-length PDE5 biosensor, is 42 amino acids shorter than the PDE5A1 isoform.”

Comment: p. 10 Comment e.: The revised Supporting Figure 1 now has the secondary structure elements identified, but the numbering of the alpha-helices (starting with α2) and beta strands should also be included for clarity.

Response: We thank the Reviewer for their valuable suggestion to include the numbering of the alpha-helices and beta strands in the revised Supporting Figure 1. In response, we have updated the figure to include this information. We agree that this addition improves the clarity of the figure.

Comment: p. 11 Comment f: In addition to correcting the typo, line 541 of the revised manuscript incorrectly describes the GAF domain as consisting of 4 alpha-helices. There are five alpha-helices in the canonical GAF domain, of which the first is missing from the crystal structure used for the simulation studies.

Response: We would like to thank the author for pointing this out. We have now edited this in the manuscript as follows:

“In the human PDE5, the GAFa domain is composed of six antiparallel β sheets (β 1- β 6) and ~~four~~ **five** α helices (α 1- α 5) with an arrangement of the secondary structural elements of $\alpha\beta\beta\beta\alpha\beta\beta\alpha$ ”

Comment: p. 14 Comment l.: Thank you for clarifying this for the reviewers, but mention of the 2K31 structure needs to be added to the manuscript as well.

Response: We appreciate the Reviewer’s comment. We would like to note that the use of the 2K31 PDB structural model is indicated on page 3 of the main text:

“Cluster of residue positions with the highest statistical coupling scores were mapped on the PDE5 GAFa domain structure (PDB: 2K31[2]).”

And on page 6 of the manuscript:

“We mapped the cluster of coevolving residues on the cGMP-bound structure of the GAFa domain of PDE5 (amino acid residues 164-312; human PDE5A1 numbering; PDB: 2K31 [2]) (Fig. 2B).”

As well as in the legends of Fig.2 and Fig. 4.:

“Fig. 2. SCA reveals a cluster of coevolving residues in GAF domains. (A) Color-coded heatmap showing pairwise scores of individual residue positions in GAF domains obtained from SCA. The positions of the nine coevolving residues are indicated. The area in the heatmap where the coevolving residue positions intersect is highlighted by a white box. (B) Cartoon representation of the GAFa domain of PDE5 (PDB: 2K31[2]) showing the coevolving cluster of residues (spheres in light blue) in the GAF domains. (C) Cartoon representation of the GAFa domain of PDE5 showing residues (side chains as sticks in pink) within a 5 Å distance from cGMP (left panel), coevolving residues (spheres in light pink; rest of the coevolving residues in light blue) within 5 Å distance from cGMP (middle panel) and coevolving residues L267 and F295 selected for further functional characterization (dark blue spheres, right panel).”

“Fig. 4. Mutation of the distant, coevolving residues alters cGMP-induced allosteric conformational change in the isolated GAFa domain of PDE5. (A) Cartoon representation of apo (PDB: 3MF0[1]) and holo (PDB: 2K31[2]) PDE5 GAFa domain structures aligned using all C α -atoms (left panel) and using C α -atom of α 2 helix (right panel) highlighting structural changes (apo: gray, holo: black) induced upon cGMP binding.”

Additionally, we have now added the following statement to the Methods section on page 3:

“The alignment of the modeled GAFa domain with its template revealed that both share similar secondary structure elements.”

Comment: p. 17 Comment #1: Please add the response to this point to the manuscript, since readers may also question whether Ca-Ca distances might change the interpretation (it doesn't).

Response: We would like to thank the author for the suggestion. We have now indicated this in the Results section of the main text on page 7:

“Similar results were obtained upon measuring the C α (C-alpha) distances between the residue positions.”

References:

1. Wang, H., H. Robinson, and H. Ke, *Conformation changes, N-terminal involvement, and cGMP signal relay in the phosphodiesterase-5 GAF domain*. The Journal of biological chemistry, 2010. **285**(49): p. 38149-38156.
2. Heikaus, C.C., et al., *Solution structure of the cGMP binding GAF domain from phosphodiesterase 5: insights into nucleotide specificity, dimerization, and cGMP-dependent conformational change*. J Biol Chem, 2008. **283**(33): p. 22749-59.

In the revised manuscript by Ahmed et al., the authors have only partially addressed the concerns raised by the three reviewers. In general, the authors' refusal to address some of the global issues about the readability and organization of the manuscript (cited by Reviewers 1 and 3) was disappointing and borderline disrespectful of the time and energy the reviewers spent providing commentary. In addition, there remain specific issues raised by the reviewers that were adequately clarified by the authors, but not incorporated into the revised manuscript.

The specific comments below reference the page number of the "Reviewers' comments" document, and the comment number/letter on each page.

p.1 Comment #1a and #1b: The refusal to reorganize the manuscript as suggested is adequately justified by the authors' responses.

p. 3-4 Comment c.: The request was for additional information on the biochemical assays, but the authors have added significant new details in the "Cell culture, transfection, and western blot analysis" section that were not requested and add further to the length of the manuscript. The information about the BRET assays, thermal stability studies, and fluorescence would have sufficed.

p. 4 Comment a.: The authors make the case that their approach of examining a single GAF domain is of greater relevance for an understanding of the entire GAF domain family. While this is indeed true, the fact is that the GAF domains found in PDE5 exist in tandem, and that the allosteric communication pathway is initiated in the cyclic nucleotide binding domain in GAFa and but is communicated through the GAFb domain to the catalytic domain. The authors need to acknowledge in the manuscript the limitations of studying only PDE5 GAFa by stating that it does not address how GAFa structure and ligand-induced conformational changes are altered when GAFb is absent.

p. 6 Comment b.: Authors have dismissed the suggestion to improve the readability of the manuscript by relegating some of the panels in Fig. 3 to Supplementary Information. Their reply is reasonable, but reflects a pattern of not recognizing the issue of "readability."

p. 6-7 Comment c.: Another example where the authors agree with the criticism, but don't alter the manuscript by eliminating the sentences that are being critiqued.

p. 8 Comment iii: Since the scientific premise for the biochemical experiments is based on the MD simulations, it is reasonable to request that at least one comparison of results between AMBER and CHARMM force fields be conducted. The authors' addition to the manuscript is acceptable, but part of the larger pattern of deflecting reasonable critiques of the work.

p. 8 Comment e.: The authors' response is in agreement with the reviewer's observation that the amplitude of the full-length protein (10-20% change) is less than for the GAFa domain (~50% change). However, the authors fail to address the much greater uncertainty in calculating the EC50 for the smaller amplitude exhibited by the full-length protein. The recommendation remains that Fig. 5E should be presented, but that Fig. 5F is of questionable value and should be omitted. [It seems odd that the WT vs L267A has a >10-fold lower p value than the WT vs F295A given the large difference in EC50 values of the two mutants relative to the WT.]

p. 9 Comment 4. Minor points, a.: The authors misinterpreted the reviewer's concern that the N-terminal region preceding the GAFa domain could influence the allosteric behavior of the PDE5 GAFa domain. PDE5A1 has additional amino acids at its N-terminus compared to PDE5A2. Wang et al (2010; ref. 16) has shown that the N-terminal sequence preceding GAFa folds over onto the GAFa domain, potentially interacting with and affecting allosteric communication. Giorgi et al. (2023) more recently reported that the N-terminal region affects the state of oligomerization of PDE5. While the reviewer understands the rationale the author use for working with the PDE5A2 sequence, the manuscript should address the possibility that the choice of isoform could have important implications for understanding the allosteric regulation of PDE5 holoenzyme.

p. 10 Comment e.: The revised Supporting Figure 1 now has the secondary structure elements identified, but the numbering of the alpha-helices (starting with $\alpha 2$) and beta-strands should also be included for clarity.

p. 11 Comment f: In addition to correcting the typo, line 541 of the revised manuscript incorrectly describes the GAF domain as consisting of 4 alpha-helices. There are five alpha-helices in the canonical GAF domain, of which the first is missing from the crystal structure used for the simulation studies.

p. 14 Comment l.: Thank you for clarifying this for the reviewers, but mention of the 2K31 structure needs to be added to the manuscript as well.

p. 17 Comment #1: Please add the response to this point to the manuscript, since readers may also question whether Ca-Ca distances might change the interpretation (it doesn't).